
# Phase transitions from heating to non-heating in $SU(1,1)$ quantum dynamics: Applications to Bose-Einstein condensates and periodically driven coupled oscillators

**Heng-Hsi Li⋆ and ⬤ Po-Yao Chang†**

Department of Physics, National Tsing Hua University, Hsinchu 30013, Taiwan

⋆ steven0823255219@gmail.com , † pychang@phys.nthu.edu.tw

## Abstract

We study the entanglement properties in non-equilibrium quantum systems with the $SU(1,1)$ structure. Through Möbius transformation, we map the dynamics of these systems following a sudden quench or a periodic drive onto three distinct trajectories on the Poincaré disc, corresponding the heating, non-heating, and a phase boundary describing these non-equilibrium quantum states. We consider two experimentally feasible systems where their quantum dynamics exhibit the $SU(1,1)$ structure: the quench dynamics of the Bose-Einstein condensates and the periodically driven coupled oscillators. In both cases, the heating, non-heating phases, and their boundary manifest through distinct signatures in the phonon population where exponential, oscillatory, and linear growths classify these phases. Similarly, the entanglement entropy and negativity also exhibit distinct behaviors (linearly, oscillatory, and logarithmic growths) characterizing these phases, respectively. Notably, for the periodically driven coupled oscillators, the non-equilibrium properties are characterized by two sets of $SU(1,1)$ generators. The corresponding two sets of the trajectories on two Poincaré discs lead to a more complex phase diagram. We identify two distinct phases within the heating region discernible solely by the growth rate of the entanglement entropy, where a discontinuity is observed when varying the parameters across the phase boundary within in heating region. This discontinuity is not observed in the phonon population.

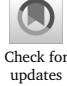

# 1  Introduction

Classifying and diagnosing quantum phases out-of-equilibrium have been broadly studied recently [1–12]. New experimental techniques including the time-dependent angle-resolved photoemission spectroscopy (trARPES) [13–17] and various pump-probe spectroscopies [18–20], have opened new doors to investigate non-equilibrium quantum phases. Numerically, the development of the time-dependent density matrix renormalization group (tDMRG) [21, 22] and quantum circuits algorithms [23–26], allow us to study the quantum dynamics of systems subjected to external drives or sudden changes of parameters (quantum quenches). Many interesting non-equilibrium phases with no equilibrium counterparts, such as time crystals [27, 28] and Floquet topological phases [1, 29], have been discovered. Despite these triumphs, analytical solutions for these systems are hard to obtain, even though many analytical techniques, such as the Bethe ansatz method [30], bosonization [31, 32], large-N treatments [33], and conformal field theory (CFT) approaches [34–36], can still be applied to non-equilibrium quantum phases. Recently, the Floquet CFT applications [37–40] to the non-equilibrium one-dimensional quantum systems at the criticality draw a huge attention due to the exact solvability. The interesting features of these Floquet critical phases are presented in the scaling properties of the entanglement entropy, which classify these non-equilibrium phases by heating, non-heating phases, and their transitions.

The Floquet CFTs utilize the algebraic structure of the $SU(1,1)$ group. While the quantum dynamics are restricted to this algebraic structure, the scaling properties of the entanglement entropy exhibit similarities to the generic quantum dynamics. For instance, in the heating phase of Floquet CFTs, the entanglement entropy exhibits linear growth over time, eventually reaching the maximally entangled state characterized by the volume law behavior of the entanglement entropy. In this paper, we focus on two experimentally feasible systems exhibiting the $SU(1,1)$ algebraic structure. However, unlike the Floquet CFTs, we do not require any conformal symmetry. By employing the $SU(1,1)$ coherent state and the Möbius transformation on the Poincaré disc (a unit disc), the heating, non-heating phases, and the phase transition can be geometrically visualized as trajectories on the Poincaré disc under the (stroboscopic) Möbius transformation.

Our first example is the $SU(1,1)$ quench dynamics of the Bose-Einstein condensates (BEC) [41–43]. By suddenly changing the interaction between bosons, stable and unstable modes can be geometrically visualized as the close and open trajectories on the Poincaré disc, as discussed in Ref. [41]. In this paper, we compute the entanglement entropy of the stable, and the unstable modes, as well as their transition. We find that the scaling properties of the entanglement entropy exhibit linearly, oscillatory, and logarithmic growths in time, respectively. These properties are identical to those of the heating, non-heating, and phase transition phases in Floquet CFTs, despite the absence of conformal symmetry in the BEC quench dynamics.

The second example involves the periodically driven coupled oscillators, where two quantum oscillators are driven by a time-dependent coupling. In one cycle of the drive, the oscillators are coupled for a $T_1$ period and then decoupled for a $T_0$ period. The quantum state evolution is described by the stroboscopic time $t = n(T_0 + T_1)$ with $n \in Z+$. Unlike Floquet CFTs and the $SU(1,1)$ quench dynamics of BEC, the periodically driven coupled oscillators possess two sets of $SU(1,1)$ structures, leading to two sets of trajectories on two Poincaré discs. These two Poincaré discs reveal more interesting non-equilibrium phases that have not been discovered in previous studies. We find there are two types of heating phases, where the exponential growth of the phonon population across these phases shows no transition. However, the growth rate of the entanglement entropy across the phase boundary exhibits a discontinuity, indicating a phase transition. This example demonstrates how entanglement measures can provide more information than other physical observables.

This paper is organized as follows: In Sec. 2, we provide the introduction of the $SU(1,1)$ algebra and the Möbius transformation on the Poincaré disc. In Sec. 3, we review the computations of entanglement entropy, the Rényi entropies, and entanglement negativity for two-mode squeezed states and the covariance matrix method. In Sec. 4, we first review geometric visualization of the BEC revival phase and the unstable mode on the Poincaré disc studied in Ref. [41]. We then demonstrate the scaling properties of the entanglement entropy can detect both the revival and the unstable phases of BEC quench dynamics. In Sec. 5, we study the periodically driven coupled oscillators described by two Poincaré discs and demonstrate that these phases can have distinct features of the scaling properties of the average phonon numbers and of entanglement entropies. Interestingly, we demonstrate there are two types of heating phases which can only be diagnosed by entanglement measures and the trace of the Möbius transformation. In addition, we numerically show the equivalence between negativity and the half-Rényi entropy for this two-mode Gaussian state. In Sec. 6. we conclude our results and give a brief discussion.

## 2 $SU(1,1)$ algebra, Möbius transformation, and Poincaré disc

The $SU(1,1)$ group is a special unitary group defined as

$$SU(1,1) = \left\{ \begin{pmatrix} \alpha & \beta \\ \beta^\star & \alpha^\star \end{pmatrix} \in GL(2,\mathbb{C}) \; : \; |\alpha|^2 - |\beta|^2 = 1 \right\}. \tag{1}$$

For the general expression above, the matrix can be parameterized by linear combinations of the three generators $\hat{K}_i, i = 0, 1, 2$, satisfying the commutation relation,

$$[\hat{K}_0, \hat{K}_1] = i\hat{K}_2, \quad [\hat{K}_1, \hat{K}_2] = -i\hat{K}_0, \quad [\hat{K}_2, \hat{K}_0] = i\hat{K}_1. \tag{2}$$

This algebra has a Fock space representation with $\hat{K}_\pm = \hat{K}_1 \pm i\hat{K}_2$, which is unitary and infinite dimensional due to its noncompactness [44]

$$\begin{aligned} \hat{K}^2 |k, m\rangle &= k(k-1)|k, m\rangle, & \hat{K}_+ |k, m\rangle &= \sqrt{(m+1)(m+2k)}|k, m+1\rangle, \\ \hat{K}_0 |k, m\rangle &= (m+k)|k, m\rangle, & \hat{K}_- |k, m\rangle &= \sqrt{m(m+2k-1)}|k, m-1\rangle, \end{aligned} \tag{3}$$

where $k$ is the Bargmann index determined by the Casimir invariant $\hat{K}^2 = \hat{K}_0^2 - (\hat{K}_+\hat{K}_- + \hat{K}_-\hat{K}_+)/2$, and $m \in \mathbb{Z}$ is the ladder index. This unitary representation will be used throughout this article. Notice that the definition (1) is a 2-dimensional representation of the group, with the generators $\hat{K}_0 = \hat{\sigma}_0/2$ and $\hat{K}_\pm = i\hat{\sigma}_\pm$.

Based on this algebra, the Hermitian Hamiltonians we are going to study throughout the article can be expressed as compositions of the generators, in which the corresponding evolution operator can be expressed as

$$\hat{U} = e^{a_+\hat{K}_+ + a_-\hat{K}_- + a_0\hat{K}_0}. \tag{4}$$

The coefficients $a_0$, $a_\pm$ are generally pure imaginary. Following Ref. [45], the above expression can be decomposed as $\hat{U} = e^{A_+\hat{K}_+}e^{\ln(A_0)\hat{K}_0}e^{A_-\hat{K}_-}$ where the parameters are transformed according to

$$\begin{aligned} \phi &= \sqrt{(a_0/2)^2 - a_+a_-}, \\ A_\pm &= \frac{(a_\pm/\phi)\sinh(\phi)}{(\cosh\phi) - (a_0/2\phi)\sinh(\phi)}, \\ A_0 &= \left(\frac{1}{(\cosh\phi) - (a_0/2\phi)\sinh(\phi)}\right)^2. \end{aligned} \tag{5}$$

By considering the lowest normalized state $|k,m\rangle = |k,0\rangle$, the above evolution operator is simplified to $\hat{U} = (A_0)^k e^{A_+\hat{K}_+}$ and operates as the $SU(1,1)$ displacement operator $e^{\xi\hat{K}_+ - \xi^\star\hat{K}_-}$ with $A_+$ being related to the displacement $\xi$ by $A_+ = \frac{\xi}{|\xi|}\tanh|\xi|$, which generates the corresponding $SU(1,1)$ generalized coherent state [46],

$$|k,A_+\rangle = (1 - |A_+|^2)^k \sum_{n=0}^{\infty} \frac{(A_+\hat{K}_+)^n}{n!}|k,0\rangle = e^{\xi\hat{K}_+ - \xi^\star\hat{K}_-}|k,0\rangle. \tag{6}$$

The expectation values of each generator with respect to the $SU(1,1)$ coherent state (CS) are directly determined by the coefficient $z = A_+$ and the Bargmann index $k$

$$\langle\hat{K}_0\rangle = k\frac{1 + |z|^2}{1 - |z|^2}, \quad \langle\hat{K}_1\rangle = \frac{2k\,\mathrm{Re}(z)}{1 - |z|^2}, \quad \langle\hat{K}_2\rangle = \frac{2k\,\mathrm{Im}(z)}{1 - |z|^2}. \tag{7}$$

This implies that the information of the $SU(1,1)$ CS is encoded in $z$, and each $SU(1,1)$ elements (operators) play the roles on how $z = A_+ \in \mathcal{D}$ evolves on the Poincaré disc (PD) $\mathcal{D}$, defined as the complex unit disc $|z| \le 1$. One can prove that the $SU(1,1)$ elements (1) exactly form Möbius transformations (MT) $\mathcal{M} \in SU(1,1)$ on the PD [38,47]

$$\mathcal{M}\cdot z = \frac{\alpha z + \beta}{\beta^\star z + \alpha^\star} = z' \in \mathcal{D}. \tag{8}$$

The MT for the $SU(1,1)$ generators can be derived by exponentiating their Pauli matrix representations [48]:

$$e^{-i\hat{K}_0\theta} = e^{-i\hat{\sigma}_z\theta/2} = \begin{pmatrix} e^{-i\theta/2} & 0 \\ 0 & e^{i\theta/2} \end{pmatrix}, \quad e^{-i\hat{K}_1\theta} = e^{\hat{\sigma}_x\theta/2} = \begin{pmatrix} \cosh(\theta/2) & \sinh(\theta/2) \\ \sinh(\theta/2) & \cosh(\theta/2) \end{pmatrix}. \tag{9}$$

The combination of the above two generators $\hat{K}_0$ and $\hat{K}_1$ leads to

$$e^{-i(\hat{K}_0+\hat{K}_1)\theta} = \begin{pmatrix} 1 + i\theta/2 & -\theta/2 \\ -\theta/2 & 1 - i\theta/2 \end{pmatrix}. \tag{10}$$

In general, the MT $\mathcal{M}$ can be written in terms of the normal form [38]

$$\frac{z_n - \gamma_-}{z_n - \gamma_+} = \eta^n\left(\frac{z_0 - \gamma_-}{z_0 - \gamma_+}\right), \tag{11}$$

where $\eta$ is some multiplier determined by the given MT, with the corresponding fixed points on the complex plane are solved by (8) using $z = z' = \gamma_\pm \in \mathcal{D}$

$$\gamma_\pm = \frac{\alpha - \alpha^\star \pm \sqrt{\text{Tr}(\mathcal{M})^2 - 4}}{2\beta^\star} . \tag{12}$$

To classify fixed points, we analyze $\text{Tr}(\mathcal{M})$ to distinguish three Möbius transformation classes:

1. Elliptic class ($\text{Tr}(\mathcal{M}) < 2$): The elliptic phase refers to $\text{Tr}(\mathcal{M}) < 2$, or by the discriminant $\Delta = \text{Tr}(\mathcal{M})^2 - 4 < 0$, which the two fixed points being

$$\gamma_\pm = \frac{\alpha - \alpha^\star \pm i\sqrt{|\Delta|}}{2\beta^\star} . \tag{13a}$$

One of them lies outside the disc, while the other remains inside. Accordingly, the multiplier $\eta$ can be represented by the phase factor $\phi \in \mathbb{R}$,

$$\eta = \frac{\text{Tr}(\mathcal{M}) + i\sqrt{|\Delta|}}{\text{Tr}(\mathcal{M}) - i\sqrt{|\Delta|}} = e^{i\phi} . \tag{13b}$$

When the system evolves from the ground state $z_0 = 0$, the above expression is simplified

$$z_n = \frac{(1 - \eta^n)\gamma_+\gamma_-}{\gamma_+ - \eta^n\gamma_-} = \frac{2\beta}{(\alpha - \alpha^\star) + \cot\left(\frac{n\phi}{2}\right)\sqrt{|\Delta|}} . \tag{13c}$$

2. Hyperbolic class ($\text{Tr}(\mathcal{M}) > 2$): The hyperbolic case refers to $\text{Tr}(\mathcal{M}) > 2$ or $\Delta > 0$, and the fixed points are

$$\gamma_\pm = \frac{\alpha - \alpha^\star \pm \sqrt{\Delta}}{2\beta^\star} . \tag{14a}$$

In comparison to the elliptic case, the two different fixed points stay on the disc boundary $|\gamma_\pm| = 1$, and the multiplier $\eta$ can be mapped to an exponential factor $e^{\phi'} \in \mathbb{R}$

$$\eta = \frac{\text{Tr}(\mathcal{M}) + \sqrt{\Delta}}{\text{Tr}(\mathcal{M}) - \sqrt{\Delta}} = e^{\phi'} . \tag{14b}$$

When the normal form starts from the ground state $z_0 = 0$, the expression is

$$z_n = \frac{(1 - \eta^n)\gamma_+\gamma_-}{\gamma_+ - \eta^n\gamma_-} = \frac{2\beta}{(\alpha - \alpha^\star) + \coth\left(\frac{n\phi'}{2}\right)\sqrt{\Delta}} . \tag{14c}$$

This result from the hyperbolic class can be mapped to the elliptic class by the analytical continuation $\phi' \to i\phi$.

3. Parabolic class ($\text{Tr}(\mathcal{M}) = 2$): For $\text{Tr}(\mathcal{M}) = 2$, the repeated fixed point $\gamma = (\alpha - \alpha^\star)/2\beta^\star$ satisfies $|\gamma| = 1$, and the normal form instead is given by

$$\frac{1}{z_n - \gamma} = \frac{1}{z_0 - \gamma} + n\beta^\star , \tag{15a}$$

and the normal form starting from the ground state $z_0 = 0$ is

$$z_n = \frac{-n\gamma^2\beta^\star}{1 - n\gamma\beta^\star} = \frac{-n(\alpha - \alpha^\star)^2}{[4 - 2n(\alpha - \alpha^\star)]\beta^\star} . \tag{15b}$$

One example for this case is given by (10).

In the following sections, we demonstrate that the entanglement quantities can distinguish different classes of Möbius transforms corresponding to different dynamical behavior/phases using the Poincaré disc representation of the normal form.

# 3 Entanglement measures

Entanglement is a phenomenon capturing the non-separability and correlations for the bipartition of the generic quantum states, with no classical counterpart. To quantify entanglement, a common approach is to compute the (von Neumann) entanglement entropy, which has been widely applied to study both ground state properties of quantum many-body systems and the phases of non-equilibrium quantum dynamics.

The entanglement entropy is computed as follows — Suppose the whole system is described by a density matrix $\rho$ which is composed of two subsystems $A$ and $B$. The reduced density matrix for subsystem A is determined by the partial trace over subsystem B, i.e. $\rho_A = \mathrm{Tr}_B(\rho)$. The entanglement entropy for subsystem A is

$$S_A = -\mathrm{Tr}(\rho_A \ln \rho_A). \tag{16}$$

There is another alternative entanglement measure called Rényi entropy $S_A^{(n)}$

$$S_A^{(n)} = \frac{1}{1-n} \ln \mathrm{Tr}\, \rho_A^n, \tag{17}$$

where $n \to 1$ reduces to the entanglement entropy (16). Besides entanglement and Rényi entropies, there exists another entanglement measure called the logarithmic negativity. It captures the entanglement from the perspective of the positive partial transpose (PPT) criterion on the density matrix $\rho$, and is defined as

$$\mathcal{E} = \ln ||\rho^{T_B}||, \tag{18}$$

where $||\rho^{T_B}||$ denotes the trace norm of the total density matrix after the partial transposition with respect to subsystem B. For pure states, the logarithmic negativity is proved to be equivalent to the half-Rényi entropy $S_A^{(1/2)}$ [49]. Although entanglement measures provide deep insights into quantum systems, entanglement entropy, and the related measures are, in general, still difficult to evaluate. In this paper, we consider two examples where the calculation of entanglement entropy and logarithmic negativity can be captured by the covariance matrix, due to the Gaussian property of the reduced density matrix. Based on this, we demonstrate that the properties of these entanglement measures reveal the dynamical properties of non-equilibrium quantum systems under a sudden quench or periodic drive.

## 3.1 Two-mode squeezed state

The first example we considered is the entanglement entropy for two-mode squeezed state with $\boldsymbol{k}$ and $-\boldsymbol{k}$ bipartition, which in general can be written as:

$$|z\rangle = \sqrt{1-|z|^2} \sum_n z^n |n\rangle_{\boldsymbol{k}} |n\rangle_{-\boldsymbol{k}}, \tag{19}$$

where $z$ corresponds to the Poincaré disc parameter. Then, the density matrix of the total system is given by

$$\hat{\rho} = |z\rangle\langle z| = (1-|z|^2) \sum_{n_{j_1},n_{k_1}} z^{n_{j_1}} (z^\star)^{n_{k_1}} \left|n_{j_1}\right\rangle_{\boldsymbol{k}} \left\langle n_{k_1}\right|_{\boldsymbol{k}} \otimes \left|n_{j_1}\right\rangle_{-\boldsymbol{k}} \left\langle n_{k_1}\right|_{-\boldsymbol{k}}. \tag{20}$$

By considering the subsystem with momentum $\boldsymbol{k}$, the reduced density matrix $\hat{\rho}_{\boldsymbol{k}}$ is the partial trace over degrees of freedom of momentum $-\boldsymbol{k}$

$$\hat{\rho}_{\boldsymbol{k}} = \mathrm{Tr}_{-\boldsymbol{k}}(|z\rangle\langle z|) = (1-|z|^2) \sum_n |z|^{2n} |n\rangle_{\boldsymbol{k}} \langle n|_{\boldsymbol{k}}. \tag{21}$$

The reduced density matrix above is diagonal in the Fock basis of the subsystem with momentum $\boldsymbol{k}$, and the entanglement entropy $S_A$ is determined by the eigenspectrum $\lambda_n = (1-|z|^2)|z|^{2n}$,

$$S_{\boldsymbol{k}} = -\sum_n \lambda_n \ln(\lambda_n) = \frac{1}{1-|z|^2}\ln\left(\frac{1}{1-|z|^2}\right) - \frac{|z|^2}{1-|z|^2}\ln\left(\frac{|z|^2}{1-|z|^2}\right). \tag{22}$$

Moreover, following the derivation details provided by [49], we derive the logarithmic negativity formula in terms of $z$ and demonstrate the equivalence between the logarithmic negativity and half-Rényi entropy. Based on the total density matrix (20), the partial transpose over degrees of freedom with momentum $-\boldsymbol{k}$ takes the form

$$\hat{\rho}^{T_B} = (1-|z|^2)\sum_{n_{j_1},n_{k_1}} z^{n_{j_1}}(z^\star)^{n_{k_1}}\left|n_{j_1}\right\rangle_{\boldsymbol{k}}\left\langle n_{k_1}\right|_{\boldsymbol{k}} \otimes \left|n_{k_1}\right\rangle_{-\boldsymbol{k}}\left\langle n_{j_1}\right|_{-\boldsymbol{k}}. \tag{23}$$

The n-th power reads

$$(\hat{\rho}^{T_B})^n = (1-|z|^2)^n \sum_{\substack{n_{j_1},\cdots,n_{j_n} \\ n_{k_1},\cdots,n_{k_n}}} z^{n_{j_1}}(z^\star)^{n_{k_1}} z^{n_{j_2}}(z^\star)^{n_{k_2}}\cdots z^{n_{j_n}}(z^\star)^{n_{k_n}}$$
$$\times \left|n_{j_1}\right\rangle_{\boldsymbol{k}}\left\langle n_{k_n}\right|_{\boldsymbol{k}} \otimes \left|n_{k_1}\right\rangle_{-\boldsymbol{k}}\left\langle n_{j_n}\right|_{-\boldsymbol{k}} \delta_{j_1,k_2}\delta_{j_2,k_1}\cdots\delta_{j_{n-1},k_n}\delta_{j_n,k_{n-1}}. \tag{24}$$

The result splits into two cases for $n$ being even ($n_e$) or odd ($n_o$)

$$(\hat{\rho}^{T_B})^n = \begin{cases} (1-|z|^2)^{n_o}\sum_{j_1,k_1}(|z|^{n_{j_1}}|z|^{n_{k_1}})^{n_o}\left|n_{j_1}\right\rangle_{\boldsymbol{k}}\left\langle n_{k_1}\right|_{\boldsymbol{k}} \otimes \left|n_{k_1}\right\rangle_{-\boldsymbol{k}}\left\langle n_{j_1}\right|_{-\boldsymbol{k}}, \\[2mm] (1-|z|^2)^{n_e}\sum_{j_1,k_1}(|z|^{n_{j_1}}|z|^{n_{k_1}})^{n_e}\left|n_{j_1}\right\rangle_{\boldsymbol{k}}\left\langle n_{j_1}\right|_{\boldsymbol{k}} \otimes \left|n_{k_1}\right\rangle_{-\boldsymbol{k}}\left\langle n_{k_1}\right|_{-\boldsymbol{k}}. \end{cases} \tag{25}$$

The trace is given by

$$\mathrm{Tr}\left(\hat{\rho}^{T_B}\right)^n = \begin{cases} \dfrac{(1-|z|^2)^{n_o}}{1-|z|^{2n_o}} = \mathrm{Tr}\left(\hat{\rho}_{-\boldsymbol{k}}^{n_o}\right), \\[4mm] \dfrac{(1-|z|^2)^{n_e}}{(1-|z|^{n_e})^2} = \mathrm{Tr}\left(\hat{\rho}_{-\boldsymbol{k}}^{n_e/2}\right)^2. \end{cases} \tag{26}$$

The above formula shows the equivalence between the partial transpose density matrix and reduced density matrix $\hat{\rho}_{-\boldsymbol{k}}$ which is given by partial trace over degrees of freedom momentum $\boldsymbol{k}$ for the whole density matrix

$$\hat{\rho}_{-\boldsymbol{k}} = \mathrm{Tr}_{\boldsymbol{k}}(|z\rangle\langle z|) = (1-|z|^2)\sum_n |z|^{2n}|n\rangle_{-\boldsymbol{k}}\langle n|_{-\boldsymbol{k}}. \tag{27}$$

One can check the trivial case where $n_o = 1$ that $\mathrm{Tr}\left(\rho^{T_B}\right) = 1$. For the case $n_e \to 1$, the analytical continuation shows the equivalence between logarithmic negativity and the half-Rényi entropy. The general expression for logarithmic negativity in terms of the Poincaré disc parameter $z$ for two-mode squeezed states is

$$\mathcal{E} = 2\ln\mathrm{Tr}\left(\rho_{-\boldsymbol{k}}^{1/2}\right) = \ln\left(\frac{1-|z|^2}{(1-|z|)^2}\right). \tag{28}$$

### 3.2 Covariance matrix method and two-mode Gaussian state

Another well-known example is the entanglement quantities of the Bosonic Gaussian state [49–51]. In this case, the entanglement entropy is directly captured by the covariance matrix $\boldsymbol{\sigma}$ [51], which is composed of first- and second-order correlation functions of the total system,

$$\sigma_{ij} = \frac{1}{2}\langle \hat{X}_i\hat{X}_j + \hat{X}_j\hat{X}_i\rangle - \langle \hat{X}_i\rangle\langle \hat{X}_j\rangle, \tag{29}$$

where $\hat{X} = (\hat{q}_1, \cdots, \hat{q}_i, \hat{p}_1, \cdots, \hat{p}_i)$ is the continuous variable vector for the system. Suppose that there are $k$ continuous variables for subsystem A, and $i - k$ continuous variables for B. The entanglement entropy for subsystem A is determined by the reduced covariance matrix $\boldsymbol{\sigma}_A$, which we trace out the matrix element relevant to subsystem B, and the symplectic eigenvalues $\mu_k$ are defined as

$$\{\mu_1, \cdots \mu_k\} = \pm i \times \text{spec}\left\{\boldsymbol{\sigma}_A \begin{pmatrix} 0 & \mathbb{I}_k \\ -\mathbb{I}_k & 0 \end{pmatrix}\right\}. \tag{30}$$

Following Ref. [50, 51], the trace power of the reduced density matrix is

$$\text{Tr}\left(\rho_A^n\right) = \prod_{i=1}^{k}\left[\left(\mu_i + \frac{1}{2}\right)^n - \left(\mu_i - \frac{1}{2}\right)^n\right]^{-1}. \tag{31}$$

From the analytical continuation, the entanglement entropy $S_A$ for the subsystem A is

$$S_A = \sum_{i=1}^{k}\left(\mu_i + \frac{1}{2}\right)\ln\left(\mu_i + \frac{1}{2}\right) - \left(\mu_i - \frac{1}{2}\right)\ln\left(\mu_i - \frac{1}{2}\right). \tag{32}$$

Additionally, for the case of the two-mode Gaussian state which is discussed in this paper. The $4 \times 4$ covariance matrix has following form [51]

$$\boldsymbol{\sigma}_{ij} = \begin{pmatrix} A & B \\ B^T & C \end{pmatrix}, \tag{33}$$

where the half-Rényi entropy can be derived by (31),

$$S_A^{(1/2)} = 2\ln\left[\left(\mu + \frac{1}{2}\right)^{1/2} - \left(\mu - \frac{1}{2}\right)^{1/2}\right]. \tag{34}$$

For the logarithmic negativity, by defining an invariant $\Delta(\boldsymbol{\sigma}) = \det(A) + \det(C) - 2\det(B)$, the alternative symplectic eigenvalues $\nu_{\pm}$ for the partial transpose covariance matrix is

$$\nu_{\pm} = \sqrt{\frac{\Delta(\boldsymbol{\sigma}) \mp \sqrt{\Delta(\boldsymbol{\sigma}) - 4\det(\boldsymbol{\sigma})}}{2}}. \tag{35}$$

The trace norm for the density matrix under the partial transpose with respect to the subsystem B [49] is given by

$$||\rho_A^{T_B}|| = \left[\left|\nu_- + \frac{1}{2}\right| - \left|\nu_- - \frac{1}{2}\right|\right]^{-1}\left[\left|\nu_+ + \frac{1}{2}\right| - \left|\nu_+ - \frac{1}{2}\right|\right]^{-1}. \tag{36}$$

Finally, one can determine the logarithmic negativity by the previous formula (17). However, unlike the previous case, it is not obvious to show the equivalence between half-Rényi entropy and the logarithmic negativity from the above expression. Based on this brief review of entanglement measures, we apply these entanglement quantities to characterize different phases of the system within the Poincaré disc framework in the following sections.

## 4 Single Poincaré disc: BEC dynamics

In this section, we first review the BEC quenching dynamics mentioned in Ref. [41]. The phase characterization of this system is based on $SU(1,1)$ algebra on the single Poincaré disc. In addition, we calculate the entanglement entropy for different BEC quench protocols, and show that it resembles the results of Floquet CFT.

## 4.1 BEC dynamics and $SU(1,1)$ algebraic structure

In the experiment of the "revival BEC" [52,53], a trapped BEC was initially prepared, and it was observed that non-zero momentum excitation number $n_k$ increases exponentially over time as the s-wave scattering length varied sinusoidally. However, such increasing behavior turned into a decrease in excitation number as they suddenly turned the s-wave scattering length by a $\pi$ phase. In Ref. [42], they explained this experimental phenomenon by the "Many-body echo" with the following Floquet-Bogoliubov Hamiltonian $\hat{H}_{\text{Bog}}$,

$$\hat{H}_{\text{Bog}} = \frac{g(t)N^2}{2V} + \sum_{k \neq 0}\left(\hat{H}_{\text{bog}}^k(t) - \frac{\epsilon_k + g(t)n_0}{2}\right), \tag{37}$$

using the perturbation theory under high frequency (low energy) limit, i.e. $1/\omega >> 1$, with $g(t) = \frac{4\pi\hbar^2}{m}a_s(t)$ denotes the scattering length in the experiment and the single-particle energy is $\epsilon_k = \hbar^2 k^2/2m$ under the uniform condensate limit. Here the non-zero momentum part $\hat{H}_{\text{bog}}^k(t)$ is

$$\hat{H}_{\text{bog}}^k(t) = 2[\epsilon_k + g(t)n_0]\hat{K}_0 + g(t)n_0(\hat{K}_+ + \hat{K}_-), \tag{38}$$

where $n_0 = |\Psi_0|^2$ is the initial BEC wave function modulus which is assumed to be independent of time during the derivation of (37), and the Hamiltonian is combined of three $SU(1,1)$ generators, $\hat{K}_0 = \frac{1}{2}(\hat{c}_k^\dagger\hat{c}_k + \hat{c}_{-k}\hat{c}_{-k}^\dagger)$, $\hat{K}_1 = \frac{1}{2}(\hat{c}_k^\dagger\hat{c}_{-k}^\dagger + \hat{c}_k\hat{c}_{-k})$, and $\hat{K}_2 = \frac{1}{2i}(\hat{c}_k^\dagger\hat{c}_{-k}^\dagger - \hat{c}_k\hat{c}_{-k})$. Similar expressions can be found in Ref. [41], where the authors rewrite the Hamiltonian as

$$\hat{H}_{\text{bog}}^k(t) = \xi_0(\boldsymbol{k})\hat{K}_0 + \xi_1(\boldsymbol{k})\hat{K}_1 + \xi_2(\boldsymbol{k})\hat{K}_2, \tag{39}$$

where the invariant effective strength $\xi = \sqrt{\xi_0^2 - \xi_1^2 - \xi_2^2}$ is related to the interaction strength $\tilde{U}$ and energy $\epsilon_k$ as $\xi_0(\boldsymbol{k}) = 2(\epsilon_k + g(t)n_0)$, $\xi_1(\boldsymbol{k}) = 2\,\text{Re}(U)$, and $\xi_2(\boldsymbol{k}) = 2\,\text{Im}(U)$ with $U = g(t)n_0$. Notice that according to (37), different momenta $\boldsymbol{k}$ modes are decoupled in the BEC quench dynamics.

Similar to the previous studies, we also start with the BEC state, i.e. the excitation vacuum state of (39) $|\Psi(0)\rangle = |0\rangle_{\boldsymbol{k}}|0\rangle_{-\boldsymbol{k}}$ with the Bargmann index $k = 1/2$. Due to the dilute limit, different $k$ modes are decoupled and we simply consider a fixed $k$ mode for the quench dynamics. The evolution of the condensate becomes

$$|z\rangle = e^{-i(\xi_0\hat{K}_0 + \xi_1\hat{K}_1 + \xi_2\hat{K}_2)t}|0\rangle_{\boldsymbol{k}}|0\rangle_{-\boldsymbol{k}} = \sqrt{1-|z|^2}\sum_n z^n |n\rangle_{\boldsymbol{k}}|n\rangle_{-\boldsymbol{k}}, \tag{40}$$

the evolution of the state can be viewed as the two-mode squeezed state with time-dependent CS parameter $z = z(t)$ determined by $\xi_0, \xi_1, \xi_2$ as

$$z(t) = -i\frac{(\xi_1 - i\xi_2)\sin(\xi t/2)}{\xi\cos(\xi t/2) + i\xi_0\sin(\xi t/2)}. \tag{41}$$

On the other hand, the above state can be written in the form of Möbius transformation (8) with $\alpha = \cos(\xi t/2) - i(\xi_0/\xi)\sin(\xi t/2)$ and $\beta = -i(\xi_1 - i\xi_2)/\xi \times \sin(\xi t/2)$ using the $SU(1,1)$ decomposition formula (5). The different parameters determine different trajectories on the Poincaré disc as shown in Fig. 1 (A). These three different trajectories correspond to different scaling behaviors of several physical quantities as shown in Figs. 1 (B), (C), and (D).

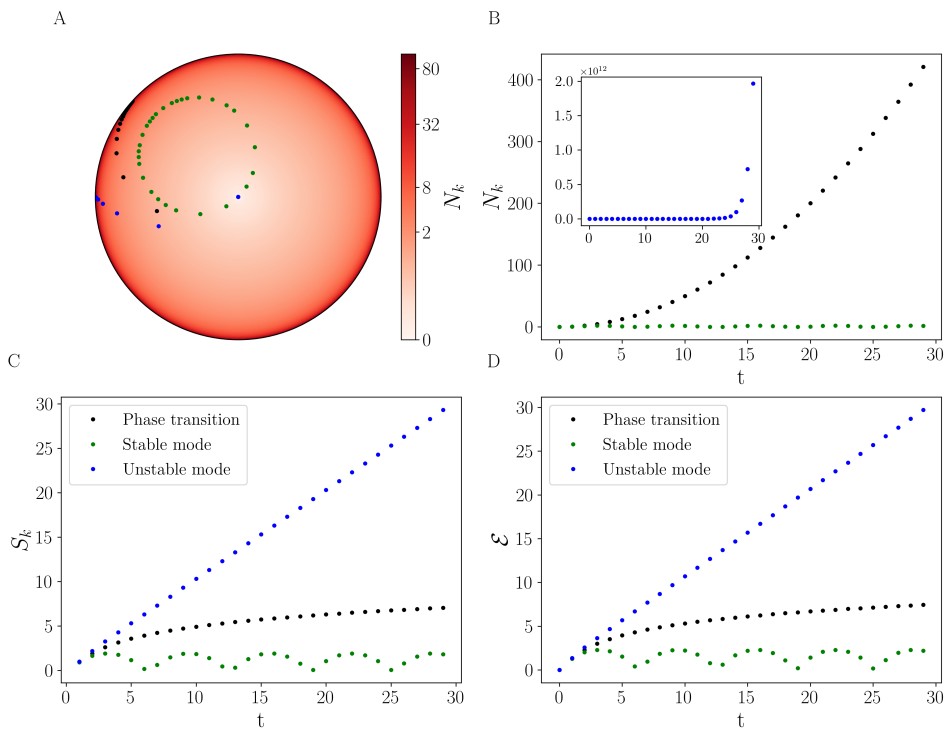

Figure 1: The quench dynamics of BEC. Here we set $\xi_1 = \xi_2 = 1$ and $\xi_0 = 1, \sqrt{2}, \sqrt{3}$ for blue(unstable), black(phase transition) and green(stable) dots respectively [41]. (A) The trajectories on the Poincaré disc describe the evolution of the BEC state and the phonon population $n_k$ distribution on the Poincaré disc. (B) The phonon population $n_k$ as a function of time. (C) Entanglement entropy $S_k$ as a function of time. (D) Logarithmic negativity $\mathcal{E}$ as a function of time.

## 4.2 Characterization of dynamical phases by the trajectories on the Poincaré disc

We first investigate the excitation $n_k$ of the BEC state, which grows as a function of time under different effective strengths $\xi$. By assuming $n_k = n_{-k}$, we can express

$$n_k = \langle \hat{c}_k^\dagger \hat{c}_k \rangle = \frac{1}{2}\frac{1+|z|^2}{1-|z|^2} - \frac{1}{2} = \frac{\xi_1^2 + \xi_2^2}{\xi^2}|\sin(\xi t/2)|^2 \propto \begin{cases} e^{ct}, & \text{unstable mode } (\xi^2 < 0), \\ ct^2, & \text{phase transition } (\xi^2 = 0), \\ \sin^2(ct), & \text{stable mode } (\xi^2 > 0), \end{cases}$$

$$(42)$$

where $c$ is a constant related to effective strength. The effective strength determines the dynamics of these phases. The BEC state remains stable and revives for $\xi^2 > 0$ and the BEC state is destroyed by the dominance of excitations for $\xi^2 < 0$. These properties can be shown by the trajectories on the Poincaré disc and from the Möbius transformation perspective. The trace condition of Möbius classes $|\text{Tr}(\mathcal{M})| = 2|\cos(\xi t/2)|$ has a one-to-one correspondence with the three cases mentioned above. Here, to be more precise, we provide an example of the evolution of states characterized by the trajectories on the Poincaré disc with the number of excitations shown in Figs. 1 (A) and (B).

For the two-mode squeezed state, we compute the entanglement entropy of the post-quench states by Eq. (41). The three phases also exhibit different growth rates of the entanglement entropy as

$$S_k \propto \begin{cases} t, & \text{unstable mode } (\xi^2 < 0), \\ \ln(t), & \text{phase transition } (\xi^2 = 0), \\ \ln(\alpha \cos(t) + \beta), & \text{stable mode } (\xi^2 > 0). \end{cases} \tag{43}$$

In this two-mode squeezed state, the entanglement entropy (22) can be expressed as the function of excitations in the BEC quenching dynamics, i.e. $S_k = (n_k + 1)\ln(n_k + 1) - n_k \ln n_k$. In summary, when the BEC state becomes unstable, the number of excitations $n_k$ increases exponentially, ultimately destroying the BEC state. At this point, the assumption made for the effective Hamiltonian (39) no longer holds as $n_k \gg 1$. On the other hand, the oscillatory behavior of $n_k$ indicates that the BEC state revives whenever $n_k \sim 0$. The stable and unstable dynamics, along with the transition between them, are characterized by the trace of the Möbius representation of the evolution operator or the sign of $\xi^2$. These properties are also reflected in the growth of the entanglement entropy (see Fig. 1 (C)). This conclusion is similar to [37, 38, 40], where different phases are characterized based on the trace of MT on a single Poincaré disc. Additionally, we present the logarithmic negativity in Fig. 1 (D).

The characterization of the dynamical properties of the BEC can be explained by the instability arising from the effective strength, which is the only parameter that distinguishes the dynamical properties of the BEC quenching dynamics. In the following example, we discuss the periodically driven coupled oscillators, where the dynamical properties depend on the driving periods. This Floquet protocol yields results similar to those of the periodically critical chain by the sine-square deformation, as discussed in Ref. [38], which requires the $SU(1,1)$ structure within the framework of conformal field theory. However, in the periodically driven coupled oscillators, conformal symmetry is not required. Instead of a single set of $SU(1,1)$ generators, this model features two distinct sets of $SU(1,1)$ generators, leading to richer dynamical properties. Notably, we observe distinct dynamical phases that can only be diagnosed from the entanglement properties but not from the phonon population.

# 5  Two Poincaré disc: Periodically driven coupled oscillators

In this section, we introduce a periodically driven system of coupled oscillators which is described by two Poincaré discs, and discuss how the phonon population and the scaling of entanglement measures distinguish the phases.

## 5.1  Preliminaries and setup

We consider two coupled oscillators periodically driven by the Hamiltonian,

$$\hat{H} = \begin{cases} \hat{H}_1 = \frac{\hat{p}_1^2}{2m} + \frac{\hat{p}_2^2}{2m} + \frac{1}{2}m\omega^2\hat{q}_1^2 + \frac{1}{2}m\omega^2\hat{q}_2^2 + C\hat{q}_1\hat{q}_2, & t - nT \in (0, T_1), \\ \hat{H}_0 = \frac{\hat{p}_1^2}{2m} + \frac{\hat{p}_2^2}{2m} + \frac{1}{2}m\omega^2\hat{q}_1^2 + \frac{1}{2}m\omega^2\hat{q}_2^2, & t - nT \in (T_1, T), \end{cases} \tag{44}$$

where $C$ is the coupling strength between two oscillators. The initial state $|G\rangle = |0\rangle_1 |0\rangle_2$ is the ground state of $\hat{H}_0$, and evolves under $\hat{H}_1$ in time $T_1$ before switching back to $\hat{H}_0$ for time $T_0$. These two steps have the total periodicity $T = T_0 + T_1$.

The Hamiltonian (44) can be decoupled by the canonical transformation $\hat{q}'_{1,2} = \frac{1}{\sqrt{2}}(\hat{q}_1 \pm \hat{q}_2)$ and $\hat{p}'_{1,2} = \frac{1}{\sqrt{2}}(\hat{p}_1 \pm \hat{p}_2)$,

$$\hat{H} = \begin{cases} \hat{H}_1 = \frac{\hat{p}'^2_1}{2m} + \frac{\hat{p}'^2_2}{2m} + \frac{1}{2}m\Omega_1^2\hat{q}'^2_1 + \frac{1}{2}m\Omega_2^2\hat{q}'^2_2, & t - nT \in (0, T_1), \\ \hat{H}_0 = \frac{\hat{p}'^2_1}{2m} + \frac{\hat{p}'^2_2}{2m} + \frac{1}{2}m\omega^2\hat{q}'^2_1 + \frac{1}{2}m\omega^2\hat{q}'^2_2, & t - nT \in (T_1, T), \end{cases} \tag{45}$$

where the decoupled frequencies become $\Omega_{1(2)}^2 = \omega^2 \pm C/m$. Also, we specify the subsystem A(B) as the oscillator $(q_{1(2)}, p_{1(2)})$. By applying the second quantization formalism, the coupled frame creation and annihilation operators are

$$\hat{a}_i = \sqrt{\frac{m\omega}{2}}\left(\hat{q}_i + \frac{i}{m\omega}\hat{p}_i\right), \qquad \hat{a}_i^\dagger = \sqrt{\frac{m\omega}{2}}\left(\hat{q}_i - \frac{i}{m\omega}\hat{p}_i\right), \tag{46}$$

and for the decoupled frame are

$$\hat{b}_i = \sqrt{\frac{m\omega}{2}}\left(\hat{q}'_i + \frac{i}{m\omega}\hat{p}'_i\right), \qquad \hat{b}_i^\dagger = \sqrt{\frac{m\omega}{2}}\left(\hat{q}'_i - \frac{i}{m\omega}\hat{p}'_i\right). \tag{47}$$

With the above definitions, the Hamiltonian under the decoupled frame can be expressed as

$$\hat{H} = \begin{cases} \hat{H}_1 = \sum_{i=1}^{2}\left((\omega + U_i)\left(\hat{b}_i^\dagger\hat{b}_i + \frac{1}{2}\right) + \frac{U_i}{2}\left((\hat{b}_i^\dagger)^2 + (\hat{b}_i)^2\right)\right), & t - nT \in (0, T_1), \\ \hat{H}_0 = \sum_{i=1}^{2}\omega\left(\hat{b}_i^\dagger\hat{b}_i + \frac{1}{2}\right), & t - nT \in (T_1, T), \end{cases} \tag{48}$$

where we assign the index $i$ as the $i$-th decoupled frame $(q'_i, p'_i)$ with the sudden change of their frequencies $\omega \to \omega + U_i$ with $U_i = \pm C/(2m\omega)$.

The algebraic structure of this periodically driven system of coupled oscillators can be expressed by two sets of the $SU(1,1)$ generators, where generators are $\hat{K}_{0,i} = \frac{1}{2}(\hat{b}_i^\dagger\hat{b}_i + \frac{1}{2})$, $\hat{K}_{1,i} = \frac{1}{4}((\hat{b}_i^\dagger)^2 + (\hat{b}_i)^2)$, and $\hat{K}_{2,i} = \frac{1}{4i}((\hat{b}_i^\dagger)^2 - (\hat{b}_i)^2)$. The Bargmann index for each one-mode representation is $k = 1/4$ which leads to the designed ground state $|k, 0\rangle$. The Hamiltonian in terms of $SU(1,1)$ generators is

$$\hat{H} = \begin{cases} \hat{H}_1 = \sum_{i=1}^{2}\left(2(\omega + U_i)\hat{K}_{0,i} + 2U_i\hat{K}_{1,i}\right), & t - nT \in (0, T_1), \\ \hat{H}_0 = \sum_{i=1}^{2}2\omega\hat{K}_{0,i}, & t - nT \in (T_1, T). \end{cases} \tag{49}$$

which has a similar setup to the quenching BEC Hamiltonian in [41] and the sine-square deformed CFT [38]. In the following sections, we characterize the dynamical phases of this periodically driven system of coupled oscillators based on trajectories on the Poincaré disc, the Möbius transformations, and their dynamical properties.

## 5.2 One-cycle drive and Möbius classes

For the periodically driven coupled oscillators, the n-cycle evolution operator $\hat{U} = \hat{U}_0\hat{U}_1\cdots\hat{U}_0\hat{U}_1$ is represented by the Möbius transformation $(\mathcal{M}_0\mathcal{M}_1)^n = \mathcal{M}^n$, with the evolution of the CS is parameterized by $z_n$. The classes for n-cycle Möbius transformation $(n \to \infty)$ can be classified by their corresponding fixed points. These classes can be determined by the trace of the one-cycle Möbius transformation. Therefore, we first set up the conventions for the one-cycle evolution and demonstrate the details of the Möbius transformation classes for each decoupled mode. The conventions introduced here will later be used in the discussion of the dynamics and the characterization of the dynamical phases.

From the Hamiltonian described by Eq. (49), one-cycle evolution operator is divided into two steps which acts independently on two decoupled modes, $\hat{U}_1 = e^{-i\hat{H}_1 T_1}$ and $\hat{U}_0 = e^{-i\hat{H}_0 T_0}$. The ground state $|G\rangle$ under single-cycle drive is $|\psi(T)\rangle = \hat{U}_0\hat{U}_1|G\rangle$. The time-dependent state

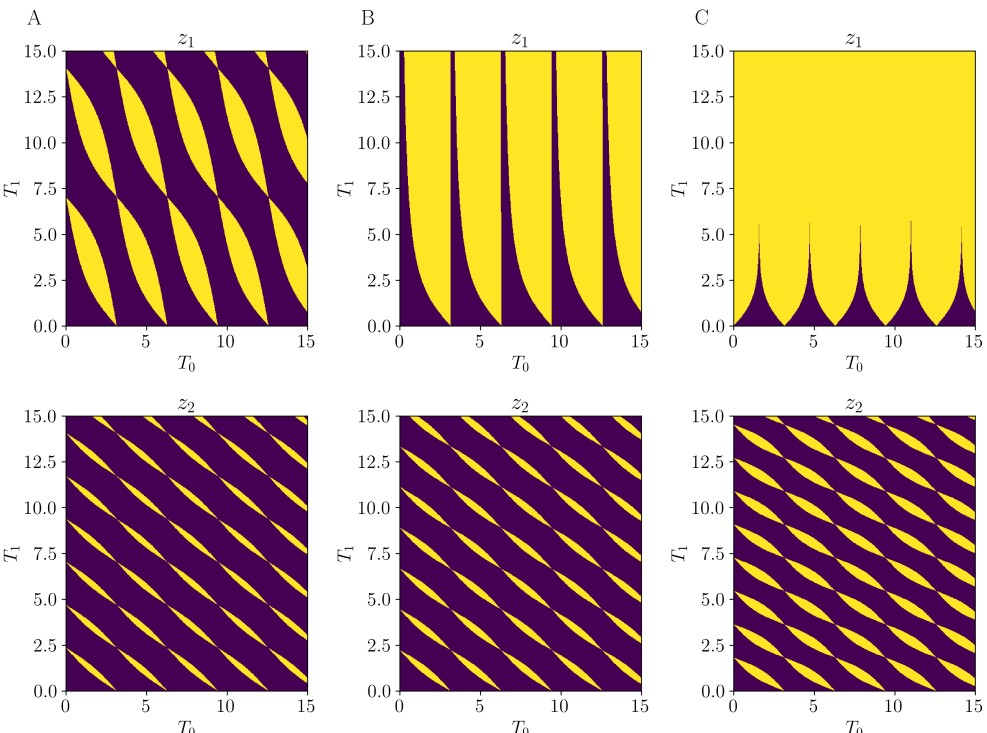

Figure 2: The trace of Móbius transformation for different phases in terms of driving period $T_0, T_1$, where the purple region corresponds to $\mathrm{Tr}(\mathcal{M}_{\mathrm{tot},i}) < 2$ (non-heating phase) and the yellow region $\mathrm{Tr}(\mathcal{M}_{\mathrm{tot},i}) > 2$ (heating phase). The phase transition $\mathrm{Tr}(\mathcal{M}_{\mathrm{tot},i}) = 2$ is captured by the intersection between the two regions. The upper panel is the $z_1$ Poincaré disc and the lower panel is the $z_2$ Poincaré disc. Here we fix $\omega = 1$ and vary the coupling constant for (a) $C = 0.8$, (b) $C = 1$ and (c) $C = 2$.

can be represented by the two independent modes which allows us to geometrize them on the two independent Poincaré discs. By focusing on the $i$-th decoupled mode, the Möbius transformation representing the evolution $\hat{U}_1$ in the $i$-th decoupled frame $\mathcal{M}_{1,i}$ has the form (8) whose $\alpha_{1,i} = \cos(\Omega_i T_1) + i \cosh(r_i) \sin(\Omega_i T_1)$ and $\beta_{1,i} = i \sinh(r_i) \sin(\Omega_i T_1)$[1] as derived in Appendix. A, where

$$\sinh(r_i) = \frac{a_{\pm,i}}{\phi_i} = \frac{-U_i}{\Omega_i}, \quad \cosh(r_i) = \frac{a_{0,i}}{2\phi_i} = -\frac{\omega + U_i}{\Omega_i}. \tag{50}$$

Here, the parameters are governed by the setup of the periodically driven coupled oscillators $a_{\pm,i} = -iU_i T_1$, $a_{0,i} = -2i(\omega + U_i)T_1$ and $\phi_i = i\sqrt{\omega^2 \pm (C/m)}\,T_1 = i\Omega_{1(2)} T_1$.

As for the $\hat{U}_0$ drive in $i$-th decoupled frame, the coefficients for Möbius transformation is $\alpha_{0,i} = e^{-i\omega T_0}$ and $\beta_{0,i} = 0$. Hence, the Möbius transformation $\mathcal{M}_{\mathrm{tot},i}$ for the entire one-cycle drive is the product of $\mathcal{M}_{0,i}$ and $\mathcal{M}_{1,i}$

$$\mathcal{M}_{\mathrm{tot},i} = \mathcal{M}_{0,i}\mathcal{M}_{1,i} = \begin{pmatrix} \alpha_{1,i} e^{-i\omega T_0} & \beta_{1,i} e^{-i\omega T_0} \\ \beta_{1,i}^{\star} e^{i\omega T_0} & \alpha_{1,i}^{\star} e^{i\omega T_0} \end{pmatrix} = \begin{pmatrix} \alpha_{\mathrm{tot},i} & \beta_{\mathrm{tot},i} \\ \beta_{\mathrm{tot},i}^{\star} & \alpha_{\mathrm{tot},i}^{\star} \end{pmatrix} \in SU(1,1). \tag{51}$$

---

[1]To make the expression simpler, the parameter $\cosh(r_i), \sinh(r_i) \in \mathbb{C}$ when $\Omega_i^2 < 0$.

The different classes of Möbius transformations are directly determined by the trace of the single-drive evolution operator $\mathcal{M}_{\text{tot},i}$,

$$\text{Tr}(\mathcal{M}_{\text{tot},i}) = 2\,\text{Re}(\alpha_1 e^{-i\omega T_0}) \begin{cases} > 2\,, & \text{hyperbolic}, \\ = 2\,, & \text{parabolic}, \\ < 2\,, & \text{elliptic}. \end{cases} \tag{52}$$

Here, the phase diagram of the periodically driven oscillator is depicted in Fig. 2. Based on the coefficients of the transformation matrices, $\text{Tr}(\mathcal{M}_{\text{tot},i})$ depends only on the ratio between two driving periods $T_1$ and $T_0$, under a constant coupling strength $C$ and natural frequencies $\omega$ for the coupled harmonic oscillators.

At this point, we have characterized different classes of Möbius transformations for single-drive evolution operators in the decoupled modes. Next, we study several physical quantities, including the phonon population and entanglement measures. For the case of the BEC quenching dynamics, these physical quantities enable us to characterize the different phases, which correspond to distinct trajectories on the two Poincare discs for this quantum dynamical system.

## 5.3  N-cycle drive and phase diagram

In the setup of the periodically driven coupled oscillators, suppose the one-cycle drive is denoted as $\mathcal{M}_{\text{tot},i}$ in the $i$-th decoupled frame, the n-cycle drive MT is then $\mathcal{M}_{\text{tot},i}^n$ and the corresponding CS parameter is denoted by $z_{n,i}$. According to the previous subsection, the evolution of the states can be fully determined by two CS parameters $z_{1,n}$ and $z_{2,n}$. In Fig. 3, the evolution of the state (subsystem A) is depicted as the trajectories on the Poincare discs, with the same driving period $T_1 = T_0$ while varying $\omega$ and $C$.

The phonon population for the subsystem A in the coupled frame $(q_1, p_1)$,

$$\langle n_A \rangle = \langle \hat{a}_1^\dagger \hat{a}_1 \rangle = \frac{1}{2}\langle \hat{b}_1^\dagger \hat{b}_1 \rangle + \frac{1}{2}\langle \hat{b}_2^\dagger \hat{b}_2 \rangle = \langle \hat{K}_{0,1} \rangle + \langle \hat{K}_{0,2} \rangle - \frac{1}{2}. \tag{53}$$

The cross term vanishes when the initial state of the system is the ground state of the oscillators in the coupled frame. This indicates that the phonon population is solely determined by the behavior of the two decoupled modes. We now summarize the three different dynamical behaviors of this driven system in terms of the classification of the Möbius transformations:

1. Elliptic class: According to the Möbius transformation (13c) and the discriminant for the $i$-th decoupled mode $\Delta_i = \text{Tr}(\mathcal{M}_{\text{tot},i})^2 - 4$, the corresponding $\langle \hat{K}_{0,i} \rangle$ is

$$\langle \hat{K}_{0,i} \rangle = \frac{1}{4}\frac{1 + |z_{\text{tot},i}|^2}{1 - |z_{\text{tot},i}|^2} = \frac{2|\beta_{\text{tot},i}|^2}{-\Delta_i}\sin^2\left(\frac{n\phi}{2}\right) + \frac{1}{4}, \tag{54}$$

where the leading coefficient is determined by the dynamical details of each decoupled mode, and the period-dependence of such expectation value is an oscillating function in terms of sine-square.

2. Hyperbolic class: According to the Möbius transformation (14c), and $\Delta_i$ as for the elliptic case,

$$\langle \hat{K}_{0,i} \rangle = \frac{1}{4}\frac{1 + |z_{\text{tot},i}|^2}{1 - |z_{\text{tot},i}|^2} = \frac{2|\beta_{\text{tot},i}|^2}{\Delta_i}\sinh^2\left(\frac{n\phi'}{2}\right) + \frac{1}{4}, \tag{55}$$

and the phonon population grows exponentially in the decoupled frame.

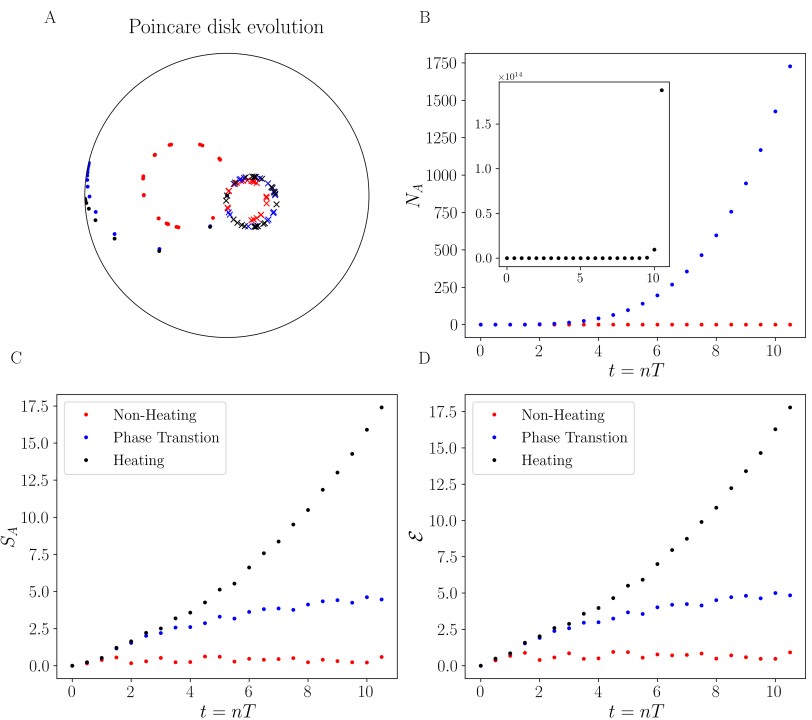

Figure 3: Non-equilibrium phases of the periodically driven coupled oscillators. Here we set $\omega = 1$ and driving periods $T_0 = T_1 = 0.25$ with different coupling strength $C = 1.5$ (non-heating phase), $C \simeq 2.01$ (logarithmic phase transition) and $C = 2.05$ (heating region I). (A) The dynamics of each case are presented by two decoupled modes. Here, we present the evolution of two independent discs together, where the dot symbol (.) stands for $z_1$ disc evolution and the cross symbol (x) for $z_2$ disc. (B) The phonon populations as functions of time for subsystem $A$. (C) Entanglement entropies as functions of time for subsystem $A$. (D) Logarithmic negativities as functions of time by taking partial transpose on subsystem $B$. The linearity of both the entanglement entropy and the negativity for heating phases in (C) and (D) can be observed after several periods.

3. Parabolic class: The Möbius transformation (15b) for the parabolic class is different from the other two classes, and $\langle \hat{K}_{0,i} \rangle$ reads

$$\langle \hat{K}_{0,i} \rangle = \frac{|\beta_{\text{tot},i}|^2}{2} n^2 + \frac{1}{4}. \tag{56}$$

It shows that the phonon population grows quadratically over time in the decoupled frame. One can further check that the expectation values of $\langle \hat{K}_{0,i} \rangle$ in this case can be obtained by taking the limit $\Delta_i \to 0$ from the other two classes.

We conclude the phonon population in subsystem A in Table 1 by using Eq. (53) and combining the results (Eqns. (54)(55)(56)) of decoupled modes. For the heating phases, the phonon population in the system grows exponentially. Conversely, in the non-heating phase, the phonon population remains oscillatory. At the phase boundary between the heating and non-heating phases, the phonon population exhibits linear growth over time. Based on Table 1,

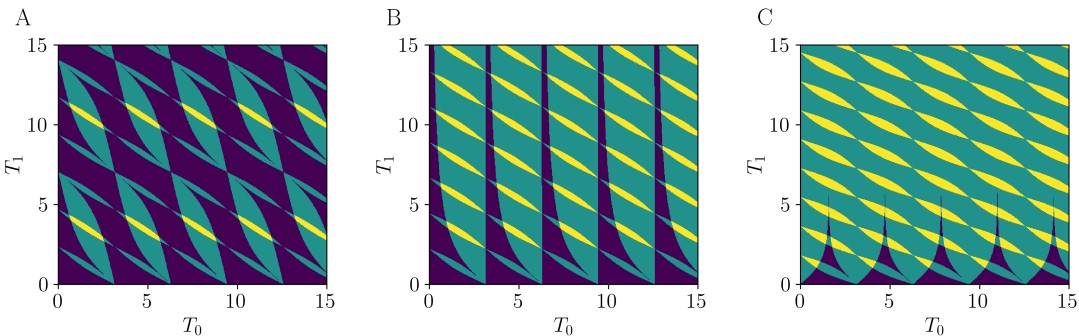

Figure 4: The phase diagram for the periodically driven coupled oscillators, where purple is the non-heating phase, while green and yellow correspond to the heating phases (I and II) with different scalings. The phase transition between the heating and non-heating phases is characterized by the intersection of the purple and green/yellow regions, while the phase boundaries between the heating phases I and II are marked by the intersection of the green and yellow regions. Here we fix $\omega = 1$ and vary the coupling constant for (a) $C = 0.8$, (b) $C = 1$ and (c) $C = 2$ as shown in Fig. 2 by overlaying the top panel and bottom panel together.

we plot the phase diagram of subsystem A for periodically driven system of coupled oscillators with different phases characterized by distinct scaling properties of the phonon population as shown in Fig. 4. These phases are also determined by the trace of Möbius transformations of each decoupled mode. In the next subsection, we explore these phases using entanglement entropy and demonstrate that the scaling properties of the phonon population alone are insufficient for complete phase characterizations.

## 5.4 Entanglement entropy and logarithmic negativity

Similar to the BEC quenching dynamics, entanglement measures such as entanglement entropy and logarithmic negativity can characterize the phases for the periodically driven coupled oscillators. As we will show below, the entanglement measures provide finer distinctions between different dynamical phases which cannot be diagnosed from the the phonon population.

Table 1: The scaling of the phonon population in subsystem A. $k$ is a proportional constant between contributions from the two discs. $c_1$ and $c_2$ depend on the dynamical details. Compared to Figure 4, the elliptic/elliptic case refers to the non-heating region (purple), the hyperbolic/elliptic case refers to the heating region I (green), and the hyperbolic/hyperbolic case refers to the heating region II (yellow). As for the cases with one parabolic class, they are identified as the different intersections between different phases (non-heating phases, heating region I, and heating region II).

| $\mathrm{Tr}(\mathcal{M}_2)$ \\ $\mathrm{Tr}(\mathcal{M}_1)$ | elliptic ($< 2$) | parabolic ($= 2$) | hyperbolic ($> 2$) |
|---|---|---|---|
| elliptic ($< 2$) | $\sin^2(c_1 t)$ | $c_1 t^2 + k\sin^2(c_2 t)$ | $\sinh^2(c_1 t) + k\sin^2(c_2 t)$ |
| parabolic ($= 2$) | $c_1 t^2 + k\sin^2(c_2 t)$ | $c_1 t^2$ | $\sinh^2(c_1 t) + kc_2 t^2$ |
| hyperbolic ($> 2$) | $\sinh^2(c_1 t) + k\sin^2(c_2 t)$ | $\sinh^2(c_1 t) + kc_2 t^2$ | $\sinh^2(c_1 t)$ |

The entanglement measures are determined by the total covariance matrix $\boldsymbol{\sigma}_{ij}$ (29), which is composed of the continuous variables of the periodically driven coupled oscillators:

$$
\begin{aligned}
\langle q_1^2 \rangle = \langle q_2^2 \rangle &= \frac{1}{2}\left(\langle q_1'^2 \rangle + \langle q_2'^2 \rangle\right) = \omega\left(\langle \hat{K}_{0,1} \rangle + \langle \hat{K}_{1,1} \rangle + \langle \hat{K}_{0,2} \rangle + \langle \hat{K}_{1,2} \rangle\right), \\
\langle p_1^2 \rangle = \langle p_2^2 \rangle &= \frac{1}{2}\left(\langle p_1'^2 \rangle + \langle p_2'^2 \rangle\right) = \frac{1}{\omega}\left(\langle \hat{K}_{0,1} \rangle - \langle \hat{K}_{1,1} \rangle + \langle \hat{K}_{0,2} \rangle - \langle \hat{K}_{1,2} \rangle\right), \\
\langle q_1 p_1 \rangle = \langle q_2 p_2 \rangle &= \frac{1}{2}\left(\langle q_1' p_1' \rangle + \langle q_2' p_2' \rangle\right) = -\langle \hat{K}_{2,1} \rangle - \langle \hat{K}_{2,2} \rangle, \\
\langle q_1 q_2 \rangle &= \frac{1}{2}\left(\langle q_1'^2 \rangle - \langle q_2'^2 \rangle\right) = \omega\left(\langle \hat{K}_{0,1} \rangle + \langle \hat{K}_{1,1} \rangle - \langle \hat{K}_{0,2} \rangle - \langle \hat{K}_{1,2} \rangle\right), \\
\langle p_1 p_2 \rangle &= \frac{1}{2}\left(\langle p_1'^2 \rangle - \langle p_2'^2 \rangle\right) = \frac{1}{\omega}\left(\langle \hat{K}_{0,1} \rangle - \langle \hat{K}_{1,1} \rangle - \langle \hat{K}_{0,2} \rangle + \langle \hat{K}_{1,2} \rangle\right), \\
\langle q_1 p_2 \rangle = \langle q_2 p_1 \rangle &= \frac{1}{2}\left(\langle q_1' p_1' \rangle - \langle q_2' p_2' \rangle\right) = -\langle \hat{K}_{2,1} \rangle + \langle \hat{K}_{2,2} \rangle.
\end{aligned}
\tag{57}
$$

It can be verified that the ten independent elements in covariance matrix reduce to six due to the identical bipartition for the two-mode Gaussian state (33). For the subsystem A, the symplectic eigenvalue (30) is determined by the reduced covariance matrix $\boldsymbol{\sigma}_A = \text{Tr}_B(\boldsymbol{\sigma})$,

$$
\mu = \sqrt{\langle q_1^2 \rangle \langle p_1^2 \rangle - \langle q_1 p_1 \rangle^2} = \sqrt{2\left(\langle \hat{K}_{0,1} \rangle \langle \hat{K}_{0,2} \rangle - \langle \hat{K}_{1,1} \rangle \langle \hat{K}_{1,2} \rangle - \langle \hat{K}_{2,1} \rangle \langle \hat{K}_{2,2} \rangle\right) + \frac{1}{8}}.
\tag{58}
$$

Here we use $\langle \hat{K}_{0,1} \rangle^2 - \langle \hat{K}_{1,1} \rangle^2 - \langle \hat{K}_{2,1} \rangle^2 = 1/16$. Accordingly, the entanglement entropy (32) for subsystem A can be expressed as a function of $z_1$ and $z_2$,

$$
\begin{aligned}
S_A &= \left(\mu + \frac{1}{2}\right)\ln\left(\mu + \frac{1}{2}\right) - \left(\mu - \frac{1}{2}\right)\ln\left(\mu - \frac{1}{2}\right) \\
&\simeq \frac{1}{2}\ln\left(\mu^2 - \frac{1}{4}\right) + 1 + o\left(\frac{1}{(\mu - 1/2)^2}\right) \\
&= \frac{1}{2}\ln\left[\frac{1}{8}\left(\frac{1 + |z_1|^2}{1 - |z_1|^2}\right)\left(\frac{1 + |z_2|^2}{1 - |z_2|^2}\right) - \frac{1}{2}\frac{z_1^\star z_2 + z_2^\star z_1}{(1 - |z_1|^2)(1 - |z_2|^2)} - \frac{1}{8}\right] + 1.
\end{aligned}
\tag{59}
$$

The initial state considered is the ground state of the decoupled oscillators, with zero entanglement entropy corresponding to $z_1 = z_2 = 0$ on both Poincaré discs. From Eq. 58, the symplectic eigenvalue is $\mu = 1/2$, which leads to zero entanglement negativity. The above expression Eq. 59 applies an approximation for long-time dynamics, where $\mu \gg 1/2$. Furthermore, the second term in Eq. (59) includes in the phase difference between $z_1$ and $z_2$, which can be simplified as

$$
\frac{z_1^\star z_2 + z_2^\star z_1}{(1 - |z_1|^2)(1 - |z_2|^2)} = \left(2\langle \hat{K}_{0,1} \rangle - \frac{1}{2}\right)\left(2\langle \hat{K}_{0,2} \rangle - \frac{1}{2}\right)\left(\frac{1}{z_1 z_2^\star} + \text{c.c.}\right),
\tag{60}
$$

where c.c represents the complex conjugates. With these expressions, the scaling properties of the entanglement entropy as a function of time can be characterized into four classes.

1. Elliptic/parabolic or elliptic/hyperbolic class: Under this case, we can make another approximation that neglects the contribution from the decoupled mode in the elliptic class, i.e., $|z_{n,2}| \simeq 0$. In this case, the mode whose MT is in elliptic class is not compatible with both hyperbolic and parabolic classes, and $\langle \hat{K}_{0,2} \rangle$ oscillates as the state oscillates

around $z_0 = 0$. Therefore, the entanglement entropy is

$$S_A \simeq \frac{1}{2} \ln\left( \frac{1}{4} \frac{|z_1|^2}{1-|z_1|^2} \right) + 1$$

$$= \begin{cases} \dfrac{1}{2} \ln\left( \dfrac{|\beta_{\text{tot},1}|^2}{\Delta_1} \sinh^2\left( \dfrac{n\phi_1}{2} \right) \right) + 1, & \text{hyperbolic/elliptic}, \\[4mm] \dfrac{1}{2} \ln\left( \dfrac{|\beta_{\text{tot},1}|^2}{4} n^2 \right) + 1, & \text{parabolic/elliptic}, \end{cases} \tag{61}$$

The entanglement entropy is $\ln(t)$ for elliptic/parabolic case (phase transition) and is linear in $t$ for elliptic/hyperbolic class (heating phase I).

2. Elliptic/elliptic or hyperbolic/hyperbolic case: In this case, it is not necessary to re-calculate the first term for Eq. (59). It contains the same information as the phonon population. Therefore, the only thing we need to take care of is the phase difference term, which is

$$\frac{z_1^\star z_2 + z_2^\star z_1}{(1-|z_1|^2)(1-|z_2|^2)} = \frac{16|\beta_{\text{tot},1}|^2|\beta_{\text{tot},2}|^2}{\Delta_1 \Delta_2} \sin^2\left( \frac{n\phi_1}{2} \right) \sin^2\left( \frac{n\phi_2}{2} \right) \left( \frac{1}{z_1 z_2^\star} + \text{c.c.} \right). \tag{62a}$$

Using the information of the normal form of the Möbius transformation, the last term can be expanded as

$$\frac{1}{z_1 z_2^\star} + \text{c.c.} = \frac{\mathfrak{a} + \mathfrak{b}\cot\left( \frac{n\phi_1}{2} \right) + \mathfrak{c}\cot\left( \frac{n\phi_2}{2} \right) + \mathfrak{d}\cot\left( \frac{n\phi_1}{2} \right)\cot\left( \frac{n\phi_2}{2} \right)}{16|\beta_{\text{tot},i}|^2|\beta_{\text{tot},i}|^2}, \tag{62b}$$

where $(\mathfrak{a}, \mathfrak{b}, \mathfrak{c}, \mathfrak{d})$ are coefficients related to the dynamical details of the trajectories on the Poincaré disc, irrelevant to the number of cycles $n$

$$\begin{cases} \mathfrak{a} = -4\beta_{\text{tot},1}^\star \beta_{\text{tot},2}(\beta_{\text{tot},1} - \beta_{\text{tot},1}^\star)(\beta_{\text{tot},2} - \beta_{\text{tot},2}^\star) + \text{c.c.}, \\ \mathfrak{b} = -4\beta_{\text{tot},1}^\star \beta_{\text{tot},2}(\beta_{\text{tot},2} - \beta_{\text{tot},2}^\star)\sqrt{|\Delta_1|} + \text{c.c.}, \\ \mathfrak{c} = 4\beta_{\text{tot},1}^\star \beta_{\text{tot},2}(\beta_{\text{tot},1} - \beta_{\text{tot},1}^\star)\sqrt{|\Delta_2|} + \text{c.c.}, \\ \mathfrak{d} = 4\beta_{\text{tot},1}^\star \beta_{\text{tot},2}\sqrt{\Delta_1 \Delta_2} + \text{c.c.}, \end{cases} \tag{62c}$$

which the entanglement entropy for the elliptic/elliptic class can be derived

$$S_A \simeq \frac{1}{2} \ln\left[ \frac{|\beta_{\text{tot},1}|^2}{|\Delta_1|} \sin^2\left( \frac{n\phi_1}{2} \right) + \frac{|\beta_{\text{tot},2}|^2}{|\Delta_2|} \sin^2\left( \frac{n\phi_2}{2} \right) - 8\sin^2\left( \frac{n\phi_1}{2} \right)\sin^2\left( \frac{n\phi_2}{2} \right) \right.$$

$$\left. \times \left( \frac{\mathfrak{a} - |\beta_{\text{tot},1}|^2|\beta_{\text{tot},2}|^2 + \mathfrak{b}\cot\left( \frac{n\phi_1}{2} \right) + \mathfrak{c}\cot\left( \frac{n\phi_2}{2} \right) + \mathfrak{d}\cot\left( \frac{n\phi_1}{2} \right)\cot\left( \frac{n\phi_2}{2} \right)}{\Delta_1 \Delta_2} \right) \right] + 1. \tag{62d}$$

This expression of the entanglement entropy is an oscillating function of time and the elliptic/elliptic class refers to the non-heating phase. As for the hyperbolic/hyperbolic class, the entanglement entropy can also be given by the analytical continuation as mentioned before with the following form,

$$S_A \simeq \frac{1}{2} \ln\left[ \frac{|\beta_{\text{tot},1}|^2}{\Delta_1} \sinh^2\left( \frac{n\phi_1'}{2} \right) + \frac{|\beta_{\text{tot},2}|^2}{\Delta_2} \sinh^2\left( \frac{n\phi_2'}{2} \right) - 8\sinh^2\left( \frac{n\phi_1'}{2} \right)\sinh^2\left( \frac{n\phi_2'}{2} \right) \right.$$

$$\left. \times \left( \frac{\mathfrak{a} - |\beta_{\text{tot},1}|^2|\beta_{\text{tot},2}|^2 + \mathfrak{b}\coth\left( \frac{n\phi_1'}{2} \right) + \mathfrak{c}\coth\left( \frac{n\phi_2'}{2} \right) + \mathfrak{d}\coth\left( \frac{n\phi_1'}{2} \right)\coth\left( \frac{n\phi_2'}{2} \right)}{\Delta_1 \Delta_2} \right) \right] + 1.$$

This expression of the entanglement entropy grows exponentially over time, and the hyperbolic/hyperbolic class corresponds to the heating phase II, where both Poincaré discs are heated.

3. Parabolic/parabolic: According to Fig. 2, the parabolic/parabolic case occurs at the intersection between the phase boundary of the two Poincaré discs. Here, the phase term has the following expression

$$\frac{1}{z_1 z_2^\star} + \text{c.c.} = \frac{\mathfrak{a}\frac{1}{n^2} + \mathfrak{b}\frac{1}{n} + \mathfrak{c}}{4|\beta_{\text{tot},1}|^2 |\beta_{\text{tot},2}|^2} \,, \tag{63a}$$

where the dynamical coefficients $(\mathfrak{a}, \mathfrak{b}, \mathfrak{c}, \mathfrak{d})$ are related to normal form of the Möbius transformation,

$$\begin{cases} \mathfrak{a} = \beta_{\text{tot},1}^\star \beta_{\text{tot},2} + \text{c.c.}\,, \\ \mathfrak{b} = \beta_{\text{tot},1}^\star \beta_{\text{tot},2}\big[(\alpha_{\text{tot},1}^\star - \alpha_{\text{tot},1}) + (\alpha_{\text{tot},2} - \alpha_{\text{tot},2}^\star)\big] + \text{c.c.}\,, \\ \mathfrak{c} = (\alpha_{\text{tot},1}^\star - \alpha_{\text{tot},1})(\alpha_{\text{tot},2} - \alpha_{\text{tot},2}^\star)\beta_{\text{tot},1}^\star \beta_{\text{tot},2} + \text{c.c.} \end{cases} \tag{63b}$$

We combine the above phase term to the previous expression of the $\langle \hat{K}_0 \rangle$, the entanglement entropy is

$$\begin{aligned} S_A &\simeq \frac{1}{2}\ln\left[ \frac{|\beta_{\text{tot},1}|^2 |\beta_{\text{tot},2}|^2}{2} n^4 + \frac{|\beta_{\text{tot},1}|^2 + |\beta_{\text{tot},2}|^2}{4} n^2 - \frac{1}{8}(\mathfrak{a}n^2 + \mathfrak{b}n^3 + \mathfrak{c}n^4) \right] + 1 \\ &= \frac{1}{2}\ln\left[ \left(\frac{|\beta_{\text{tot},1}|^2 |\beta_{\text{tot},2}|^2}{2} - \frac{\mathfrak{c}}{8}\right)n^4 - \frac{\mathfrak{b}}{8}n^3 + \left(\frac{|\beta_{\text{tot},1}|^2 + |\beta_{\text{tot},2}|^2}{4} - \frac{\mathfrak{a}}{8}\right)n^2 \right] + 1 \,. \end{aligned} \tag{63c}$$

However, this case is identical to the phase transition for the parabolic/elliptic case. We examine the scaling properties of the entanglement entropy for these two cases and find there is no discontinuity. Here we simply choose the parameters $m = 1$ and $C = \omega$ with the driving period $T_0 = n\pi$ and $T_1 = 2\pi s/\Omega_2$ as $s > 0$ for the middle phase diagram of Fig. 4, which $\beta_{\text{tot},2} = 0$. Since both the elliptic/parabolic and parabolic/parabolic classes exhibit the same scaling law and are smoothly connected, they belong to the same phase transition.

4. Hyperbolic/parabolic: This case refers to the intersections between heating regions I and II on the phase diagram, where the phase term has the following expression,

$$\frac{1}{z_1 z_2^\star} + \text{c.c.} = \frac{\mathfrak{a}\frac{1}{n} + \mathfrak{b} + \mathfrak{c}\coth\left(\frac{n\phi_1}{2}\right) + \mathfrak{d}\frac{1}{n}\coth\left(\frac{n\phi_1}{2}\right)}{8|\beta_{\text{tot},1}|^2 |\beta_{\text{tot},2}|^2} \,, \tag{64a}$$

and the dynamical coefficients $(\mathfrak{a}, \mathfrak{b}, \mathfrak{c}, \mathfrak{d})$ are also determined,

$$\begin{cases} \mathfrak{a} = 2(\alpha_{\text{tot},1} - \alpha_{\text{tot},1}^\star)\beta_{\text{tot},1}^\star \beta_{\text{tot},1} + \text{c.c.}\,, \\ \mathfrak{b} = 2(\alpha_{\text{tot},1} - \alpha_{\text{tot},1}^\star)(\alpha_{\text{tot},2} - \alpha_{\text{tot},2}^\star)\beta_{\text{tot},1}^\star \beta_{\text{tot},2} + \text{c.c.}\,, \\ \mathfrak{c} = 2\beta_{\text{tot},1}^\star \sqrt{\Delta_1} + \text{c.c.}\,, \\ \mathfrak{d} = 2(\alpha_{\text{tot},2} - \alpha_{\text{tot},2}^\star)\beta_{\text{tot},2}\beta_{\text{tot},1}^\star \sqrt{\Delta_1} + \text{c.c.} \end{cases} \tag{64b}$$

The entanglement entropy is

$$\begin{aligned} S_A &\simeq \frac{1}{2}\ln\left[ \frac{2|\beta_{\text{tot},1}|^2 |\beta_{\text{tot},2}|^2}{\Delta_1} n^2 \sinh^2\left(\frac{n\phi_1}{2}\right) + \frac{4|\beta_{\text{tot},1}|^2}{\Delta_1}\sinh^2\left(\frac{n\phi_1}{2}\right) + |\beta_{\text{tot},2}|^2 n^2 \right. \\ &\quad \left. - \frac{1}{4\Delta_1}n^2 \sinh^2\left(\frac{n\phi_1}{2}\right)\left(\mathfrak{a}\frac{1}{n} + \mathfrak{b} + \mathfrak{c}\coth\left(\frac{n\phi_1}{2}\right) + \mathfrak{d}\frac{1}{n}\coth\left(\frac{n\phi_1}{2}\right)\right) \right] + 1 \,. \end{aligned} \tag{64c}$$

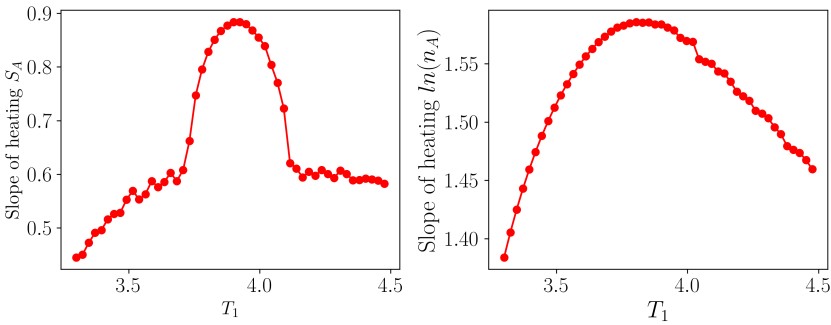

Figure 5: The slope of the linearly growth of the entanglement entropy $\alpha(T_0, T_1)$ and the exponential factors $c(T_0, T_1)$ for the phonon population $n_A \simeq e^{c(T_0,T_1)t}$ in the heating region. Here we vary the parameter $T_1$ with fixed $\omega = 1$, $C = 0.8$, and $T_0 = 1$, which corresponds to the system going from the heating phase I to the heating phase II and back to the heating phase I. The former quantity exists a jump, while the latter remains smooth as the driving period (or the trace of MT) changes. Here the parameters $\omega = 1$, $C = 0.8$, and $T_0 = 1$, with the phase transition between two different heating phases occurs at roughly $T_1 \simeq 3.72$, which matches the phase diagram in Fig. 4.

The above expression gives the linear scaling of the entanglement entropy. It indicates the heating phase I (hyperbolic/elliptic), heating phase II (hyperbolic/hyperbolic), and their phase boundary (hyperbolic/parabolic) all have the same scaling property of the entanglement entropy. However, the growth rates of the entanglement entropy for these phases are different. Thus, these phases can still be distinguished by the entanglement measures as discussed below.

Although the two heating phases (I and II) and their phase boundary exhibit the same scaling property of the entanglement entropy, they differ in the Möbius class of the decoupled mode $z_2$. To diagnose these two phases, we compute the growth rate $\alpha(T_0, T_1)$ of the entanglement entropy, expressed as $S_A = \alpha(T_0, T_1)t + \beta(T_0, T_1)$, across the phase boundaries (transitioning from heating phase I to heating phase II and back to heating phase I) as a function of $T_1$ for fixed $\text{Tr}(\mathcal{M}_{\text{tot},2})$, as shown in Fig. 4. The non-smooth behavior of $\alpha(T_0, T_1)$ at the phase boundaries can be observed.

Table 2: The scaling properties of the entanglement entropy for the long-time dynamics. $\alpha(T_0, T_1), \beta(T_0, T_1)$ are dynamical constants which depend on $T_0, T_1$. However, they are independent of $t = n(T_0 + T_1)$.

| $\text{Tr}(\mathcal{M}_2)$ \ $\text{Tr}(\mathcal{M}_1)$ | elliptic ($< 2$) | parabolic ($= 2$) | hyperbolic ($> 2$) |
|---|---|---|---|
| elliptic ($< 2$) | $\ln(\alpha\cos(t) + \beta)$ | $\ln(\alpha(T_0,T_1)t)$ | $\alpha(T_0,T_1)t + \beta(T_0,T_1)$ |
| parabolic ($= 2$) | $\ln(\alpha(T_0,T_1)t)$ | $\ln(\alpha(T_0,T_1)t)$ | $\alpha(T_0,T_1)t + \beta(T_0,T_1)$ |
| hyperbolic ($> 2$) | $\alpha(T_0,T_1)t + \beta(T_0,T_1)$ | $\alpha(T_0,T_1)t + \beta(T_0,T_1)$ | $\alpha(T_0,T_1)t + \beta(T_0,T_1)$ |

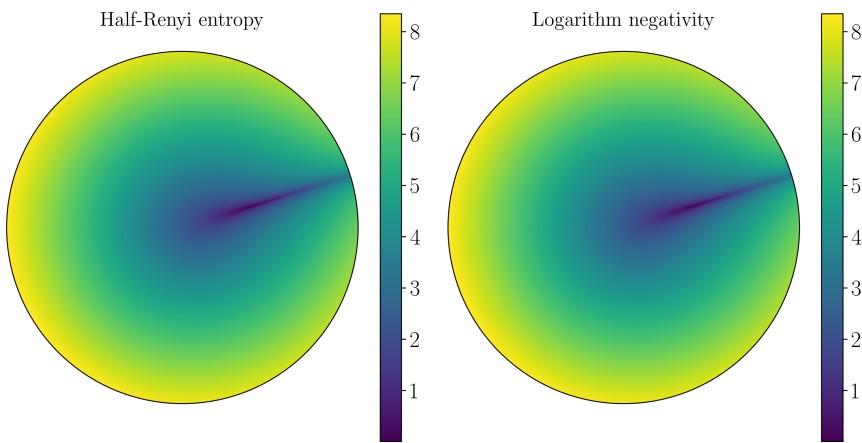

Figure 6: The logarithmic negativity (left)and the half-Rényi entropy (right) are mapped on the circle of the squeezed parameter. Here, we map the CS parameter $z_1$ to the squeezed parameter $(r, \theta)$ by $z_1 = -e^{i\theta} \tanh(|r|)$ and fix $z_2 = 0.9 \times \exp(0.5i)$. We demonstrate that the logarithmic negativity and the half-Rényi entropy are identical on the circle of the squeezed parameter for the $z_1$ decoupled mode. The cutoff of the boundary is $r = 5.5$ where $z_1 = \tanh(0.55) \simeq 0.99$ near the Poincaré disc boundary for $z_1$ decoupled mode.

In contrast, the exponential growth rate $c(T_0, T_1)$ of the phonon population $n_A = e^{c(T_0, T_1)t}$ remains a smooth function across these phase boundaries. This indicates that entanglement measures provide finer distinctions between dynamical phases compared to other physical observables. This observation demonstrates that the scaling properties of the entanglement entropy, Móbius transformation, and trajectories on the Poincaré discs characterize different dynamical phases.

Finally, we numerically compute the scaling property of the entanglement entropy of the periodically driven coupled oscillators as shown in Fig. 3 for $\text{Tr}(\mathcal{M}_{\text{tot},2}) < 2$ as a function of $\text{Tr}(\mathcal{M}_{\text{tot},1})$. We conclude the scaling properties of entanglement entropy in Table 2.

Besides entanglement entropy, the scaling property of logarithmic negativity derived from the covariance matrix (57), using the alternative symplectic eigenvalue (35) and the trace norm formula (36), is presented in Fig. 5 (D). We numerically demonstrate the equivalence between half-Rényi entropy (34) and the logarithmic negativity in Fig. 6. Although the generic proof of the equivalence between logarithmic negativity and the half-Rényi entropy for pure states is discussed in Refs. [49, 54], demonstrating this equivalence analytically for a general two-mode Gaussian state remains challenging. Furthermore, the equivalency shows that logarithmic negativity has scaling properties similar to those of entanglement entropy, as shown in Fig. 3(D).

# 6 Discussions

In this paper, we study the $SU(1, 1)$ dynamics of quantum systems through the classification of Möbius transformations, trajectories of Poincaré disc, scaling behaviors of observable quantities, and scaling properties of the entanglement entropy. We specifically demonstrate these properties in the contexts of BEC quench dynamics and periodically driven coupled oscillators. First, we show that the trace of Möbius transformation matrix $\text{Tr}(\mathcal{M})$ characterizes the phases for these non-equilibrium states, which can exhibit linear, logarithm and oscillating properties

for the entanglement entropy. Such criterion also possesses geometric visualization on the Poincaré disc, in which three phases correspond to three different fixed points and distinct trajectories. For the BEC quench dynamics, these classes of the Möbius transformation are directly captured by the instability of the system. Interestingly, for the periodically driven coupled oscillates, there exist more refined dynamical phases characterized by two sets of Möbius transformations. Specifically, the heating phases I and II cannot be distinguished from the scaling property of the phonon population. However, the growth rate of the entanglement entropy has a jump across these phases, indicating that the entanglement measures serve as a more refined tool for diagnosing these dynamical phases.

Compared to previous studies on Floquet CFT dynamics [38], although the essential structure for both their work and ours is $SU(1,1)$, our work does not require conformal symmetry and offers simpler physical realizations. The $SU(1,1)$ quench dynamics of BEC have been discussed in Ref. [41]. Furthermore, it is interesting to extend our discussion to the one-dimensional harmonic chain with a gapped spectrum. In this system, the absence of the Floquet CFT description is notable. However, based on our study of the periodically driven coupled oscillators, multiple sets of $SU(1,1)$ in the one-dimensional harmonic chain potentially should lead to a more complex phase diagram. Another interesting extension would be to include couplings beyond $SU(1,1)$. In this case, the generic state describing the entire system mixes states $|k,m\rangle$ with different Bargmann indices $k$, and one should consider hopping between different Poincaré discs, as studied in Ref. [55].

## Acknowledgments

We are grateful to Xueda Wen, Shin-Tza Wu, and Austen Lamacraft for useful discussions. P.-Y. C. thanks the National Center for Theoretical Sciences, Physics Division (Taiwan) for its support.

**Funding information** P.-Y. C. acknowledges support from the National Science and Technology Council of Taiwan under Grants No. NSTC 112-2636-M-007-007, No. 112-2112-M-007-043, and No. 113-2112-M-007-019.

## A One cycle Mobius transformation for periodic driven oscillators

In this appendix, we give a thorough derivation of the Mobius transformation representation for the evolution operator. Referring to the main text, we represent the evolution operator $\hat{U}_1$ as $\mathcal{M}_1$ while $\hat{U}_0$ as $\mathcal{M}_0$. First, as the operators in different decoupled frames commute with each other, we can separate the evolution operators and only focus on the $i$-th decoupled frame. Based on that, we have $a_{\pm,i} = -iU_i T_1$ snd $a_{0,i} = -2i(\omega + U_i)T_1$. Then, the phase $\phi_i$ is determined by

$$\phi_i = i\sqrt{\omega^2 \pm (C/m)}\, T_1 = i\Omega_{1(2)} T_1\,, \tag{A.1}$$

which is directly related to the decoupled frequencies of the two oscillators. Then on, with these coefficients, we can define a new set of hyperbolic functions in terms of $a_{\pm,i}$, $a_{0,i}$ and $\phi_i$

$$\sinh(r_i) = \frac{a_{\pm,i}}{\phi_i} = \frac{-U_i}{\Omega_i}\,, \quad \cosh(r_i) = \frac{a_{0,i}}{2\phi_i} = -\frac{\omega + U_i}{\Omega_i}\,. \tag{A.2}$$

One can check that this representation holds for $\cosh(r_i)^2 - \sinh(r_i)^2 = 1$, with $A_{\pm,i}$ and $A_{0,i}$ become

$$A_{\pm,i} = \frac{\sinh(r_i)\sinh(i\Omega_i T_1)}{\cosh(i\Omega_i T_1) - \cosh(r_i)\sinh(i\Omega_i T_1)},$$
$$A_{0,i} = \left(\frac{1}{\cosh(i\Omega_i T_1) - \cosh(r_i)\sinh(i\Omega_i T_1)}\right)^2. \tag{A.3}$$

For the ladder operators $\hat{K}_\pm = \hat{K}_1 \pm i\hat{K}_2$, the exponential map can be read as

$$e^{\hat{K}_+ \theta} = \begin{pmatrix} 1 & \theta \\ 0 & 1 \end{pmatrix}, \qquad e^{\hat{K}_- \theta} = \begin{pmatrix} 1 & 0 \\ -\theta & 1 \end{pmatrix}. \tag{A.4}$$

Conclude the calculation above by combining these given coefficients with the exponential maps of the generators, we have [56]

$$\begin{aligned}
\mathcal{M}_{1,i} &= e^{A_{+,i}\hat{K}_{+,i}} e^{\ln(A_{0,i})\hat{K}_{0,i}} e^{A_{-,i}\hat{K}_{-,i}} = \begin{pmatrix} 1 & A_{+,i} \\ 0 & 1 \end{pmatrix} \begin{pmatrix} e^{\ln(A_{0,i})/2} & 0 \\ 0 & e^{-\ln(A_{0,i})/2} \end{pmatrix} \begin{pmatrix} 1 & 0 \\ -A_{-,i} & 1 \end{pmatrix} \\
&= \begin{pmatrix} \cosh(i\Omega_i T_1) + \cosh(r_i)\sinh(i\Omega_i T_1) & \sinh(r_i)\sinh(i\Omega_i T_1) \\ -\sinh(r_i)\sinh(i\Omega_i T_1) & \cosh(i\Omega_i T_1) - \cosh(r_i)\sinh(i\Omega_i T_1) \end{pmatrix} \\
&= \begin{pmatrix} \cos(\Omega_i T_1) + i\cosh(r_i)\sin(\Omega_i T_1) & i\sinh(r_i)\sin(\Omega_i T_1) \\ -i\sinh(r_i)\sin(\Omega_i T_1) & \cos(\Omega_i T_1) - i\cosh(r_i)\sin(\Omega_i T_1) \end{pmatrix}. \tag{A.5}
\end{aligned}$$

Hence, by setting $\alpha_{1,i} = \cos(\Omega_i T_1) + i\cosh(r_i)\sin(\Omega_i T_1)$ and $\beta_{1,i} = i\sinh(r_i)\sin(\Omega_i T_1)$, the first drive $\mathcal{M}_{1,i}$ belongs to SU(1,1) group. As for the second drive $\mathcal{M}_{0,i}$, the MT is totally a Pauli spinor with an additional phase

$$\mathcal{M}_{0,i} = e^{-2i\omega \hat{K}_{0,i} T_0} = e^{-i\omega\hat{\sigma}_z T_0} = \begin{pmatrix} e^{-i\omega T_0} & 0 \\ 0 & e^{i\omega T_0} \end{pmatrix}. \tag{A.6}$$

Finally, the whole cycle of evolution is represented by $\mathcal{M}_{\text{tot},i} = \mathcal{M}_{0,i}\mathcal{M}_{1,i}$, which has the explicit form

$$\mathcal{M}_{\text{tot},i} = \begin{pmatrix} \alpha_{1,i}e^{-i\omega T_0} & \beta_{1,i}e^{-i\omega T_0} \\ \beta_{1,i}^\star e^{i\omega T_0} & \alpha_{1,i}^\star e^{i\omega T_0} \end{pmatrix} = \begin{pmatrix} \alpha_{\text{tot},i} & \beta_{\text{tot},i} \\ \beta_{\text{tot},i}^\star & \alpha_{\text{tot},i}^\star \end{pmatrix}, \tag{A.7}$$

as the convention setup in the main text.

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
