# Peer review of "Phase transitions from Heating to non-heating in SU(1, 1) quantum dynamics: applications to Bose-Einstein condensates and periodically driven coupled oscillators"

_SciPost Physics Core, doi:SciPost Phys. Core 8, 018 (2025)_

## Round 1 · Referee Report · Anonymous (Referee 1) · 2024-9-13

Report

The authors consider the entanglement measures of various systems described by SU(1,1) algebra, driven by quantum quenches or periodic protocols. I think the results are interesting and the paper deserves to be published in Scipost Physics Core after addressing my comments.

  1. It it not obvious which quantity is k dependent and which is not. For example, after Eq. (37), the xi_{0,1,2}(k) are defined, by \xi is k independent and later \xi_{0,1,2} appear without the k argument. I find this confusing here and at other places. Similarly, in Fig. 1, n_k and S_k appears without specifying the k values.

  2. For Sec. 4, there is a momentum sum for the BEC Hamiltonian, but later this sum is omitted. Is it due to the peculiar initial state?

  3. What happens if there are also other terms in the Hamiltonian which do not preserve the SU(1,1) algebraic structures. Maybe distinct the generators of distinct k modes are coupled to each other or some non-linear powers of the generators appear?

Recommendation

Ask for minor revision

---

## Round 1 · Referee Report · Anonymous (Referee 2) · 2024-11-29

Strengths

The Authors investigate the problem of bosonic coherent state dynamics under $\mathrm{SU}(1,1)$ dynamics in two physical systems, namely a Bose-Einstein condensate (BEC) and a pair of periodically driven harmonic oscillators. In both cases, they identify different phases that can be tuned by changing some parameters, and they diagnose it by studying the proliferation of excitations and entanglement. The main novelties of the work are 1) the calculation of momentum-space entanglement entropy and negativity of the BEC, and 2) the full study of the Floquet dynamics of the two oscillators under the alternating evolution of two Hamiltonians. Overall, the topic of the paper is interesting, and the results are presented clearly with step-by-step derivations of the main equations, making them relatively easy to follow.

Weaknesses

The main weakness of the manuscript is that it is not written in perfect English, and it contains multiple grammar mistakes. This does not prohibit the comprehension of the text, but the quality of writing does not currently meet the standards for a publication.

Report

All in all, I believe the problem under investigation is interesting and the results are convincing. For this reason, I am willing to recommend this article for publication after major changes have been made. In particular, I expect the Authors to significantly improve the quality of the text, as well as to address some questions and observations that I put forward in the remainder of this report. In addition, I believe the title can be misleading, as the nomenclature ``Entanglement transitions'' more commonly refers to measurement-induced phase transitions in the literature, and I thus suggest to update it.

Requested changes

Here are some questions that I would like the Authors to answer.

1- The Hamiltonian of Eq.~(36) describes a BEC with contact interactions in free space. In realistic experimental implementations, the condensate must be constrained by a trap, leading to the addition of an external potential. This changes multiple properties of the BEC. Can the Authors comment on whether or not they expect their result to apply also to a condensate in a trap? Is there a limit in which the trapped condensate Hamiltonian reduces to the free-space one considered in the manuscript? I understand that the formalism used in the article is probably no longer applicable in absence of translational invariance, but I believe that it is important to contextualize the problem in a realistic situation, especially considering that the Authors claim the system is experimentally feasible. 2- The entanglement entropy considered in Sec.~4 is evaluated in momentum space between modes with opposite momenta. Is it possible to access this quantity experimentally? If so, I believe that adding appropriate references would benefit their discussion, and would also highlight that their calculation of $S_\mathbf{k}$ is not purely of theoretical interest. 3- In many-body systems, the entanglement entropy is more commonly evaluated in real space. Can the results on $S_\mathbf{k}$ be leveraged to infer what happens in real space as well, for instance by considering a half-system bipartition? My expectation for the BEC is that the real-space entropy grows linearly in time in all phases, because even in the stable phase where $S_\mathbf{k}$ oscillates, the different $\mathrm{k}$ mode show revivals at dephased frequencies, and thus no collective revival occurs. What do the Authors think? 4- What are the practical consequences of $S_\mathbf{k}\sim t$? Does this mean that eventually the condensate phase is unstable due to proliferation of excitations? What about the stable phase and the phase transition point? I think further commenting on this can enrich the discussion. 5- Can the Authors clarify whether or not (or to what extent) the phase diagrams they observe in both systems actually depend on the choice of the initial state? I would expect that as long as the system is prepared in any coherent state, the phase diagrams should remain the same, as they are fully determined by the Möbius map. 6- Do the Authors expect that the phase diagram of the paired oscillators is robust against the addition of dissipation? What happens in the realistic situation where the system is held at finite (small) temperature?

Minor comments

Below I list some minor comments and corrections I found while reading the manuscript. I point out that I checked explicitly the correctness of all equations up to Sec.~5.2 included, with the exception of Sec.~3.2 1- At the beginning of Sec.~2, I would make it more clear that Eq.~(1) is just the 2-dimensional representation of the group, but that the operators considered later on are not in that form and are instead unitary. While this is already stated, I think it can be emphasized explicitly to avoid any confusion. 2- Providing an explicit example of $\hat{K}_i$ in terms of Pauli matrices would be useful. 3- I believe that in Eq.~(4) it should be specified that $a_\pm,a_0$ are purely imaginary. Otherwise, the same operator can also be realized using the generators of the $\mathrm{SU}(2)$ group. 4- In Eq.~(6), specify that $|k,A_+\rangle = e^{\xi \hat{K}+-\xi^*\hat{K}-}|k,0\rangle$. 5- Can the Authors provide references to the statement One can prove that the $\mathrm{SU}(2)$ elements exactly form Möbius transformations'' above Eq.~(8)? 6- Make it clear that $Tr(\mathcal{M})$ coincides with the trace of Eq.~(10). 7- The reader might not be familiar with the notion of the multiplier $\eta$ introduced in Eq.~(12b). I suggest that Eq.~(12c) is used to define it, and after explaining its meaning you later put Eq.~(12b) to provide its explicit form. 8- I feel like Eq.~(18) lacks some context. I would add a comment specifying that you consider a coherent state in momentum space involving modes $\pm\mathrm{k}$, that your goal is to evaluate momentum-space entanglement, and that these states will appear later. 9- I believe Eq.~(19) has two swapped indices. The correct version should have the index $k_1$ at the bra with momentum $\mathrm{k}$, and $j_1$ at the ket with momentum $-\mathrm{k}$. This does not impact later results. 10- I believe there is the same problem in the second line of Eq.~(24). The correct version should have the index $j_1$ at the bra with momentum $\mathrm{k}$, and $k_1$ at the ket with momentum $-\mathrm{k}$. 11- At the beginning of Sec.~4.1, the Authors should provide either an explicit derivation of the effective Hamiltonian, or references that obtain it. 12- Above Eq.~(38), $|\Psi_0\rangle$ is not simply the ground state of Eq.~(37), it is its ground state when assuming a specific parameter choice. I believe its clearer if they simply refer to it as the excitation vacuum state. 13- Referring to Eq.~(40), specify that $k=1/2$. 14- In Eq.~(40), the squared sine should also have a modulus, otherwise it can become negative for $\xi^2<0$. 15- Around Eq.~(41), the Authors should mention that in the unstable mode eventually the approximation used to derive the effective Hamiltonian eventually breaks down because $N_{k\neq 0}\gg 1$. 16- At the start of Sec.~5.2, specify that$n$-cycle'' means to apply $\mathcal{M}$ $n$ times. 17- Before Eq.~(48), write the expressions of $\sinh r_i$ and $\cosh r_i$ explicitly in the main text. 18- In the conclusions, I believe the Authors mean QED'' rather thanQCD'' cavity. Some references or discussion on how the Floquet dynamics of the oscillators could be implemented would also be useful.

Recommendation

Ask for major revision

  • validity: good
  • significance: good
  • originality: good
  • clarity: good
  • formatting: excellent
  • grammar: acceptable

Author:  Po-Yao Chang  on 2024-12-17  [id 5044]

(in reply to Report 2 on 2024-11-29)

We thank both reviewers for carefully reading through our manuscript and giving constructive comments and suggestions. Here we reply the major comments from the both reviewers.

Response to the first reviewer:

  1. It not obvious which quantity is k dependent and which is not. For example, after Eq. (37), the $\xi_{0,1,2}(k)$ are defined, by $\xi$ is k independent and later $\xi_{0,1,2}$ appear without the k argument. I find this confusing here and at other places. Similarly, in Fig. 1, $n_k$ and $S_k$ appears without specifying the k values. Response: To make it more clear, we rewrite sec.4. $\xi_{0}$ depends on momentum due to $E_{k}$. As for $\xi_{1,2}$, the dependence of momentum $k$ lies on the fact that the $U$ is given by the condensation wave function $\Psi_{0}$. Since different $\boldsymbol{k}$ modes are decoupled, we single out $\boldsymbol{k}$ mode and chosen $\xi_{0,1,2}$ to be fixed. $n_k$ and $S_k$ are the corresponding observables for the fixed $\xi_{0,1,2}$ of the given $\boldsymbol{k}$ mode.

  2. For Sec. 4, there is a momentum sum for the BEC Hamiltonian, but later this sum is omitted. Is it due to the peculiar initial state? Response: Same as the first question, we here single out $\boldsymbol{k}$ mode and $\xi_{0,1,2}$ are fixed.

  3. What happens if there are also other terms in the Hamiltonian which do not preserve the SU(1,1) algebraic structures. Maybe distinct the generators of distinct k modes are coupled to each other or some non-linear powers of the generators appear? Response: According to [arXiv preprint arXiv:2205.07429], the different Bargmann index $|k,m\rangle$ states may mixed. We've added this discussion in our last paragraph of sec.6

Response to the second reviewer:

  1. The Hamiltonian of Eq.~(36) describes a BEC with contact interactions in free space. In realistic experimental implementations, the condensate must be constrained by a trap, leading to the addition of an external potential. This changes multiple properties of the BEC. Can the Authors comment on whether or not they expect their result to apply also to a condensate in a trap? Is there a limit in which the trapped condensate Hamiltonian reduces to the free-space one considered in the manuscript? I understand that the formalism used in the article is probsably no longer applicable in absence of translational invariance, but I believe that it is important to contextualize the problem in a realistic situation, especially considering that the Authors claim the system is experimentally feasible. Answer: In the derivation of BEC quenching Hamiltonian [Phys. Rev. A 102, 011301(R)], we take the uniform condensate limit. For the shallow potential, the uniform condensate limit is still applicable. However, if we reconsider the the deep trapping potential, one should modify the dispersion $\epsilon_{k}$ is no longer to be $\hbar^{2}k^{2}/2m$ and is modified by the trapping potential. However, one can still reevaluate $\epsilon_{k}$ to be some functions of $k$ and can adapt it in to $\xi_0$. The following discussion of the heating/non-heating phases will be similar but with parameters depends on the trapping term. In the new version of the manuscript, we mention this issue in sec. 4.

  2. The entanglement entropy considered in Sec.~4 is evaluated in momentum space between modes with opposite momenta. Is it possible to access this quantity experimentally? If so, I believe that adding appropriate references would benefit their discussion, and would also highlight that their calculation of $S_{k}$ is not purely of theoretical interest. Answer: Here we consider the entanglement between the pair of bosons with opposite momenta. In the real space bipartition, one can consider the boson carrying $+k$ is moving to the right and the boson carrying $-k$ is moving to the left. I.e., these bosons can be though as EPR pairs. Thus the entanglement entropy we considered in the paper can be visualized as the entanglement entropy contributed from these EPR pairs in real space. As we discussed in the manuscript, the entanglement entropy can be evaluated experimentally by $S_{k} = (n_{k} +1)\ln(n_{k} +1) + n_{k}\ln{n_{k}}$, where the $n_k$ is the number of the excitations carrying momentum with $+k$ which can be detected from the experiments.

  3. In many-body systems, the entanglement entropy is more commonly evaluated in real space. Can the results on $S_{k}$ be leveraged to infer what happens in real space as well, for instance by considering a half-system bipartition? My expectation for the BEC is that the real-space entropy grows linearly in time in all phases, because even in the stable phase where $S_{k}$ oscillates, the different $k$ mode show revivals at dephased frequencies, and thus no collective revival occurs. What do the Authors think? Answer: Consider bipartition in the real space, we refer to the picture in [Phys. Rev. Lett. 121, 243001], where the maximal correlation (related to entanglement) reaches maximum at $\theta$ and $\theta + \pi$ pairing, resembling the pattern that $\boldsymbol{k}$ and $-\boldsymbol{k}$ is maximally entangled in our manuscript. In this case, the entanglement doesn't increase as there are no additional EPR-pair created.

  4. What are the practical consequences of $S_{k} \sim t$ Does this mean that eventually the condensate phase is unstable due to proliferation of excitations? What about the stable phase and the phase transition point? I think further commenting on this can enrich the discussion. Answer: We think that the condensate is unstable in the heating regime due to the proliferation of excitations. And for the stable phase, the condensate actually revives as one can observe by the Poincare disk (as our main text). For the phase transition, the time to reach the proliferation of excitation scales at $\log(t)$. I.e., the condensate gets destroyed at $t \to \infty$.

  5. Can the Authors clarify whether or not (or to what extent) the phase diagrams they observe in both systems actually depend on the choice of the initial state? I would expect that as long as the system is prepared in any coherent state, the phase diagrams should remain the same, as they are fully determined by the M\"obius map. Answer: In the theoretical setup, the choice of initial (coherent) state will not affect the trace argument. But one should rederive the corresponding M\"obius transformation, as the normal form of MT actually depends on $z_{0}$ (i.e. the initial choice of coherent state, where in our whole discussion is chosen to be $z_{0} = 0$). However, for the BEC case, the choice of different initial states affects the consequences, as some initial coherent states possess significant amounts of non-zero $n_{k}$ excitations that break the approximation of the Bogoliubov Hamiltonian.

  6. Do the Authors expect that the phase diagram of the paired oscillators is robust against the addition of dissipation? What happens in the realistic situation where the system is held at finite (small) temperature? Answer: As long as the dissipation term satisfying su(1,1) algebra, the argument about the trace of M\'obius transformation still holds. The heating pattern is still there with different MT arguments depending on the dissipation parameters. For the finite temperature cases, one need to reformulate the setup from a pure state to a mixed (thermal) state. From the Poincare disc point of view, the pure state is a point on the disc and we analyze the trajectory of this point on the Poincare disc. For the mixed state, we can map it as a patch or patches on the disc and analyze how the patches evolve on the disc. This situation is much more complicated and interesting. It deserves a comprehensive investigation for future study.

Here we reply to the minor comments from the second reviewer

  1. At the beginning of Sec.~2, I would make it more clear that Eq.~(1) is just the 2-dimensional representation of the group, but that the operators considered later on are not in that form and are instead unitary. While this is already stated, I think it can be emphasized explicitly to avoid any confusion. Answer: We have added it at the bottom of page 2.

  2. Providing an explicit example of $\hat{K}_i$ in terms of Pauli matrices would be useful. Answer: We have added it at the bottom of page 2.

  3. I believe that in Eq.~(4) $a_{\pm}$, $a_0$ are purely imaginary. Otherwise, the same operator can also be realized using the generators of the SU(2) group. Answer: We add a short notification under the equation (4).

  4. In Eq.~(6), specify that $|k, A_+\rangle = e^{\xi k_+- \xi^*k_-} |k,- \rangle$. Answer: We have added it in the equation (6).

  5. Can the Authors provide references to the statement One can prove that the SU(2) elements exactly form Mobius transformations'' above Eq.~(8)? Answer: We have added the references mentioning the statement.

  6. Make it clear that $Tr \mathcal{M}$coincides with the trace of Eq.~(10). Answer: We have added it under the equation (15b).

  7. The reader might not be familiar with the notion of the multiplier introduced in Eq.~(12b). I suggest that Eq.~(12c) is used to define it, and after explaining its meaning you later put Eq.~(12b) to provide its explicit form. Answer: We rewrite the paragraph at the bottom of page 4, where we first introduce the normal form, then we demonstrate the three classes corresponding to different $\eta$.

  8. I feel like Eq.~(18) lacks some context. I would add a comment specifying that you consider a coherent state in momentum space involving modes $\pm k$, that your goal is to evaluate momentum-space entanglement, and that these states will appear later. Answer: We add a line about the bipartition above the equation (19).

  9. I believe Eq.~(19) has two swapped indices. The correct version should have the index $K_1$ at the bra with momentum $k$, and $j_1$ at the ket with momentum $-k$. This does not impact later results. Answer: We corrected the swapped indices in the equation (20).

  10. I believe there is the same problem in the second line of Eq.~(24). The correct version should have the index $j_1$ at the bra with momentum $k$, and $k_1$ at the ket with momentum $-k$. Answer: We corrected the swapped indices in the equation (25).

  11. At the beginning of Sec.~4.1, the Authors should provide either an explicit derivation of the effective Hamiltonian, or references that obtain it. Answer: We rewrote the paragraph in section 4.1, and we also added references about the revival BEC and Floquet-Bogoliubov Hamiltonian.

  12. Above Eq.~(38), $|\Psi_0 \rangle$ is not simply the ground state of Eq.~(37), it is its ground state when assuming a specific parameter choice. I believe it's clearer if they simply refer to it as the excitation vacuum state. Answer: We have corrected it above the equation (40).

  13. Referring to Eq.~(40), specify that $k=1/2$. Answer: We added the short line about the Bargamann index in the paragraph above the equation (40).

  14. In Eq.~(40), the squared sine should also have a modulus, otherwise it can become negative for $\xi^2 <0$. Answer: We have corrected it in the equation (42).

  15. Around Eq.~(41), the Authors should mention that in the unstable mode eventually the approximation used to derive the effective Hamiltonian eventually breaks down because $N_{k \neq 0} \gg 1$. Answer: We added a short line in the paragraph below the equation (43).

  16. At the start of Sec.~5.2, specify that "n-cycle'' means to apply $\mathcal{M}$ n times. Answer: We rewrote the first paragraph in section 5.2 and mentioned the definition of n-cycle.

  17. Before Eq.~(48), write the expressions of $\sinh r_i$ and $\cosh r_i$ explicitly in the main text. Answer: We mention the expressions in equation (50) with some comments around that equation.

  18. In the conclusions, I believe the Authors mean QED'' rather than QCD'' cavity. Some references or discussion on how the Floquet dynamics of the oscillators could be implemented would also be useful. Answer: We deleted the comments about the QED cavity in the second paragraph of section 6.

List of Changes to article We made several changes to the text: 1. We change the title to "Phase transitions from Heating to non-heating in SU(1, 1) quantum dynamics: applications to Bose-Einstein condensates and periodically driven coupled oscillators".

  1. We add Pauli representation for SU(1,1) group in sec.2.

  2. We mentioned that the coefficients $a_{0}$, $a_{\pm}$ are generally pure imaginary below equation (4).

  3. We introduce the normal form before the listing of different cases in equation (11).

  4. We mention that equation (10) relates to the $Tr(M) = 2$ below equation (15b).

  5. We mention the bipartition we consider is the $k$ and $-k$ above equation (19).

  6. We correct the typos of the swap index in (20) and (25).

  7. We rewrite sec.4.1 by adding references to experiments and theoretical derivation. Also, we specify that we consider only the momentum $k$ state in Fig.1.

  8. In sec.4.2, we specify that the effective Hamiltonian no longer holds for $n_{k} \gg 1$.

  9. We move the introduction of 'n-cycle' from sec.5.3 to sec.5.2.

  10. We add definition of $\sinh(r_{i})$ and $\cosh(r_{i})$ in equation (50).

  11. We add discussions about adding couplings outside of SU(1,1) algebra in the last paragraph of the section.6

  12. We correct several typos and grammar, and we also rewrite several sentences for better reading.

---

## Round 2 · Author Response

We thank both reviewers for carefully reading through our manuscript and giving constructive comments and suggestions. Here we reply the major comments from the both reviewers.

Response to the first reviewer:

  1. It not obvious which quantity is k dependent and which is not. For example, after Eq. (37), the $\xi_{0,1,2}(k)$ are defined, by $\xi$ is k independent and later $\xi_{0,1,2}$ appear without the k argument. I find this confusing here and at other places. Similarly, in Fig. 1, $n_k$ and $S_k$ appears without specifying the k values. Response: To make it more clear, we rewrite sec.4. $\xi_{0}$ depends on momentum due to $E_{k}$. As for $\xi_{1,2}$, the dependence of momentum $k$ lies on the fact that the $U$ is given by the condensation wave function $\Psi_{0}$. Since different $\boldsymbol{k}$ modes are decoupled, we single out $\boldsymbol{k}$ mode and chosen $\xi_{0,1,2}$ to be fixed. $n_k$ and $S_k$ are the corresponding observables for the fixed $\xi_{0,1,2}$ of the given $\boldsymbol{k}$ mode.

  2. For Sec. 4, there is a momentum sum for the BEC Hamiltonian, but later this sum is omitted. Is it due to the peculiar initial state? Response: Same as the first question, we here single out $\boldsymbol{k}$ mode and $\xi_{0,1,2}$ are fixed.

  3. What happens if there are also other terms in the Hamiltonian which do not preserve the SU(1,1) algebraic structures. Maybe distinct the generators of distinct k modes are coupled to each other or some non-linear powers of the generators appear? Response: According to [arXiv preprint arXiv:2205.07429], the different Bargmann index $|k,m\rangle$ states may mixed. We've added this discussion in our last paragraph of sec.6

Response to the second reviewer:

  1. The Hamiltonian of Eq.~(36) describes a BEC with contact interactions in free space. In realistic experimental implementations, the condensate must be constrained by a trap, leading to the addition of an external potential. This changes multiple properties of the BEC. Can the Authors comment on whether or not they expect their result to apply also to a condensate in a trap? Is there a limit in which the trapped condensate Hamiltonian reduces to the free-space one considered in the manuscript? I understand that the formalism used in the article is probsably no longer applicable in absence of translational invariance, but I believe that it is important to contextualize the problem in a realistic situation, especially considering that the Authors claim the system is experimentally feasible. Answer: In the derivation of BEC quenching Hamiltonian [Phys. Rev. A 102, 011301(R)], we take the uniform condensate limit. For the shallow potential, the uniform condensate limit is still applicable. However, if we reconsider the the deep trapping potential, one should modify the dispersion $\epsilon_{k}$ is no longer to be $\hbar^{2}k^{2}/2m$ and is modified by the trapping potential. However, one can still reevaluate $\epsilon_{k}$ to be some functions of $k$ and can adapt it in to $\xi_0$. The following discussion of the heating/non-heating phases will be similar but with parameters depends on the trapping term. In the new version of the manuscript, we mention this issue in sec. 4.

  2. The entanglement entropy considered in Sec.~4 is evaluated in momentum space between modes with opposite momenta. Is it possible to access this quantity experimentally? If so, I believe that adding appropriate references would benefit their discussion, and would also highlight that their calculation of $S_{k}$ is not purely of theoretical interest. Answer: Here we consider the entanglement between the pair of bosons with opposite momenta. In the real space bipartition, one can consider the boson carrying $+k$ is moving to the right and the boson carrying $-k$ is moving to the left. I.e., these bosons can be though as EPR pairs. Thus the entanglement entropy we considered in the paper can be visualized as the entanglement entropy contributed from these EPR pairs in real space. As we discussed in the manuscript, the entanglement entropy can be evaluated experimentally by $S_{k} = (n_{k} +1)\ln(n_{k} +1) + n_{k}\ln{n_{k}}$, where the $n_k$ is the number of the excitations carrying momentum with $+k$ which can be detected from the experiments.

  3. In many-body systems, the entanglement entropy is more commonly evaluated in real space. Can the results on $S_{k}$ be leveraged to infer what happens in real space as well, for instance by considering a half-system bipartition? My expectation for the BEC is that the real-space entropy grows linearly in time in all phases, because even in the stable phase where $S_{k}$ oscillates, the different $k$ mode show revivals at dephased frequencies, and thus no collective revival occurs. What do the Authors think? Answer: Consider bipartition in the real space, we refer to the picture in [Phys. Rev. Lett. 121, 243001], where the maximal correlation (related to entanglement) reaches maximum at $\theta$ and $\theta + \pi$ pairing, resembling the pattern that $\boldsymbol{k}$ and $-\boldsymbol{k}$ is maximally entangled in our manuscript. In this case, the entanglement doesn't increase as there are no additional EPR-pair created.

  4. What are the practical consequences of $S_{k} \sim t$ Does this mean that eventually the condensate phase is unstable due to proliferation of excitations? What about the stable phase and the phase transition point? I think further commenting on this can enrich the discussion. Answer: We think that the condensate is unstable in the heating regime due to the proliferation of excitations. And for the stable phase, the condensate actually revives as one can observe by the Poincare disk (as our main text). For the phase transition, the time to reach the proliferation of excitation scales at $\log(t)$. I.e., the condensate gets destroyed at $t \to \infty$.

  5. Can the Authors clarify whether or not (or to what extent) the phase diagrams they observe in both systems actually depend on the choice of the initial state? I would expect that as long as the system is prepared in any coherent state, the phase diagrams should remain the same, as they are fully determined by the M\"obius map. Answer: In the theoretical setup, the choice of initial (coherent) state will not affect the trace argument. But one should rederive the corresponding M\"obius transformation, as the normal form of MT actually depends on $z_{0}$ (i.e. the initial choice of coherent state, where in our whole discussion is chosen to be $z_{0} = 0$). However, for the BEC case, the choice of different initial states affects the consequences, as some initial coherent states possess significant amounts of non-zero $n_{k}$ excitations that break the approximation of the Bogoliubov Hamiltonian.

  6. Do the Authors expect that the phase diagram of the paired oscillators is robust against the addition of dissipation? What happens in the realistic situation where the system is held at finite (small) temperature? Answer: As long as the dissipation term satisfying su(1,1) algebra, the argument about the trace of M\'obius transformation still holds. The heating pattern is still there with different MT arguments depending on the dissipation parameters. For the finite temperature cases, one need to reformulate the setup from a pure state to a mixed (thermal) state. From the Poincare disc point of view, the pure state is a point on the disc and we analyze the trajectory of this point on the Poincare disc. For the mixed state, we can map it as a patch or patches on the disc and analyze how the patches evolve on the disc. This situation is much more complicated and interesting. It deserves a comprehensive investigation for future study.

Here we reply to the minor comments from the second reviewer

  1. At the beginning of Sec.~2, I would make it more clear that Eq.~(1) is just the 2-dimensional representation of the group, but that the operators considered later on are not in that form and are instead unitary. While this is already stated, I think it can be emphasized explicitly to avoid any confusion. Answer: We have added it at the bottom of page 2.

  2. Providing an explicit example of $\hat{K}_i$ in terms of Pauli matrices would be useful. Answer: We have added it at the bottom of page 2.

  3. I believe that in Eq.~(4) it should be specified that $a_\pm$, $a_0 are purely imaginary. Otherwise, the same operator can also be realized using the generators of the SU(2) group. Answer: We add a short notification under the equation (4).

  4. In Eq.~(6), specify that $|k, A_{\pm}\ranlge = e^{\xi K_+ - \xi^*- K-} | k, 0 \rangle$. Answer: We have added it in the equation (6).

  5. Can the Authors provide references to the statement One can prove that the SU(2) elements exactly form Mobius transformations'' above Eq.~(8)? Answer: We have added the references mentioning the statement.

  6. Make it clear that $Tr \mathcal{M}$ coincides with the trace of Eq.~(10). Answer: We have added it under the equation (15b).

  7. The reader might not be familiar with the notion of the multiplier introduced in Eq.~(12b). I suggest that Eq.~(12c) is used to define it, and after explaining its meaning you later put Eq.~(12b) to provide its explicit form. Answer: We rewrite the paragraph at the bottom of page 4, where we first introduce the normal form, then we demonstrate the three classes corresponding to different $\eta$.

  8. I feel like Eq.~(18) lacks some context. I would add a comment specifying that you consider a coherent state in momentum space involving modes $\pm k$, that your goal is to evaluate momentum-space entanglement, and that these states will appear later. Answer: We add a line about the bipartition above the equation (19).

  9. I believe Eq.~(19) has two swapped indices. The correct version should have the index $k_1$ at the bra with momentum $k$, and $j_1$ at the ket with momentum $-k$. This does not impact later results. Answer: We corrected the swapped indices in the equation (20).

  10. I believe there is the same problem in the second line of Eq.~(24). The correct version should have the index $j_1$ at the bra with momentum $k$, and $k_1$ at the ket with momentum $-k$. Answer: We corrected the swapped indices in the equation (25).

  11. At the beginning of Sec.~4.1, the Authors should provide either an explicit derivation of the effective Hamiltonian, or references that obtain it. Answer: We rewrote the paragraph in section 4.1, and we also added references about the revival BEC and Floquet-Bogoliubov Hamiltonian.

  12. Above Eq.~(38), $|\Psi \rangle$ is not simply the ground state of Eq.~(37), it is its ground state when assuming a specific parameter choice. I believe it's clearer if they simply refer to it as the excitation vacuum state. Answer: We have corrected it above the equation (40).

  13. Referring to Eq.~(40), specify that $k=\1/2$. Answer: We added the short line about the Bargamann index in the paragraph above the equation (40).

  14. In Eq.~(40), the squared sine should also have a modulus, otherwise it can become negative for $\xi^2 < 0$. Answer: We have corrected it in the equation (42).

  15. Around Eq.~(41), the Authors should mention that in the unstable mode eventually the approximation used to derive the effective Hamiltonian eventually breaks down because $N_{k \neq 0} \gg 1$. Answer: We added a short line in the paragraph below the equation (43).

  16. At the start of Sec.~5.2, specify that "n-cycle'' means to apply $\mathcal{M}$ n times. Answer: We rewrote the first paragraph in section 5.2 and mentioned the definition of n-cycle.

  17. Before Eq.~(48), write the expressions of $\sinh r_i$ and $\cosh r_i$ explicitly in the main text. Answer: We mention the expressions in equation (50) with some comments around that equation.

  18. In the conclusions, I believe the Authors mean QED'' rather than QCD'' cavity. Some references or discussion on how the Floquet dynamics of the oscillators could be implemented would also be useful. Answer: We deleted the comments about the QED cavity in the second paragraph of section 6.

---

## Round 2 · List of Changes

We made several changes to the text: 1. We change the title to "Phase transitions from Heating to non-heating in SU(1, 1) quantum dynamics: applications to Bose-Einstein condensates and periodically driven coupled oscillators".

  1. We add Pauli representation for SU(1,1) group in sec.2.

  2. We mentioned that the coefficients $a_{0}$, $a_{\pm}$ are generally pure imaginary below equation (4).

  3. We introduce the normal form before the listing of different cases in equation (11).

  4. We mention that equation (10) relates to the $Tr(M) = 2$ below equation (15b).

  5. We mention the bipartition we consider is the $k$ and $-k$ above equation (19).

  6. We correct the typos of the swap index in (20) and (25).

  7. We rewrite sec.4.1 by adding references to experiments and theoretical derivation. Also, we specify that we consider only the momentum $k$ state in Fig.1.

  8. In sec.4.2, we specify that the effective Hamiltonian no longer holds for $n_{k} \gg 1$.

  9. We move the introduction of 'n-cycle' from sec.5.3 to sec.5.2.

  10. We add definition of $\sinh(r_{i})$ and $\cosh(r_{i})$ in equation (50).

  11. We add discussions about adding couplings outside of SU(1,1) algebra in the last paragraph of the section.6

  12. We correct several typos and grammar, and we also rewrite several sentences for better reading.

---

## Editorial Decision

published